

# Inversion of geothermal heat flux in a thermomechanically coupled nonlinear Stokes ice sheet model

Hongyu Zhu[1], Noemi Petra[2], Georg Stadler[3], Tobin Isaac[4], Thomas J.R. Hughes[1,5], and Omar Ghattas[1,6,7]

[1]Institute for Computational Engineering & Sciences, The University of Texas at Austin, Austin, TX 78712, USA
[2]Applied Mathematics, School of Natural Sciences, University of California, Merced, CA 95343, USA
[3]Courant Institute of Mathematical Sciences, New York University, New York, NY 10012, USA
[4]Computation Institute, University of Chicago, Chicago, IL 60637, USA
[5]Department of Aerospace Engineering & Engineering Mechanics, The University of Texas at Austin, Austin, TX 78712, USA
[6]Jackson School of Geosciences, The University of Texas at Austin, Austin, TX 78712, USA
[7]Department of Mechanical Engineering, The University of Texas at Austin, Austin, TX 78712, USA

*Correspondence to:* Hongyu Zhu (zhuhongyu@ices.utexas.edu)

**Abstract.** We address the inverse problem of inferring the basal geothermal heat flux from surface velocity observations using an instantaneous thermomechanically coupled nonlinear Stokes ice flow model. This is a challenging inverse problem since the map from basal heat flux to surface velocity observables is indirect: the heat flux is a boundary condition for the thermal advection-diffusion equation, which couples to the nonlinear Stokes ice flow equations, which then determine the surface ice flow velocity. This multiphysics inverse problem is formulated as a nonlinear least-squares optimization problem with a cost functional that includes the data misfit between surface velocity observations and model predictions. A Tikhonov regularization term is added to render the problem well-posed. We derive adjoint-based gradient and Hessian expressions for the resulting PDE-constrained optimization problem and propose an inexact Newton method for its solution. As a consequence of the Petrov-Galerkin discretization of the energy equation, we show that discretization and differentiation do not commute; that is, the order in which we discretize the cost functional and differentiate it affects the correctness of the gradient. Using two and three-dimensional model problems, we study the prospects for and limitations of the inference of the geothermal heat flux field from surface velocity observations. The results show that the reconstruction improves as the noise level in the observations decreases, and that small wavelength variations in the geothermal heat flux are difficult to recover. We analyze the ill-posedness of the inverse problem as a function of the number of observations by examining the spectrum of the Hessian of the cost functional. Motivated by the popularity of operator-split or staggered solvers for forward multiphysics problems—i.e., those that drop two-way coupling terms to yield a one-way coupled forward Jacobian—we study the effect on the inversion of a one-way coupling of the adjoint energy and Stokes equations. We show that taking such a one-way coupled approach for the adjoint equations can lead to an incorrect gradient and premature termination of optimization iterations due to loss of a descent direction stemming from inconsistency of the gradient with the contours of the cost functional. Nevertheless, one may still obtain a reasonable approximate inverse solution particularly if important features of the reconstructed solution emerge early in optimization iterations, before the premature termination.



# 1 Introduction

We consider the following multiphysics inverse problem: given surface velocity observations and a non-Newtonian full Stokes ice sheet flow model governed by thermomechanically coupled mass, momentum, and energy equations, infer the unknown basal geothermal heat flux field. Grid-based discretization of the basal heat flux field leads to a high-dimensional inverse problem. The main aim of this paper is to present an efficient method for solving this large-scale coupled-physics inverse problem and to use model problems to study the prospects for, and limitations of, inferring the geothermal heat flux from surface ice velocities.

Ice sheet models are characterized by unknown or uncertain parameters stemming from the lack of direct observations of the interior and the base of the ice sheet. Unknown parameters include those that represent basal friction, basal topography, rheology, geothermal heat flux, and ice thickness. The geothermal heat flux parameter field, in particular, has a strong influence on the thermal state of the ice, and hence plays a critical role in understanding the dynamics of the ice sheet through its effect on basal and internal ice temperatures (Fahnestock et al., 2001; Maule et al., 2005; Petrunin et al., 2013). Estimates of this parameter field have been obtained via inference using global seismic tomographic models (Shapiro and Ritzwoller, 2004), satellite magnetic data models (Maule et al., 2005), or tectonic models (Pollack et al., 1993). However, these inferred basal heat flux fields do not agree with one another in large regions. While sensitivity studies show that the resulting uncertainties in the geothermal heat flux have an impact on the ice flow, especially in the slow flow regions (Pollard et al., 2005; Larour et al., 2012), to the best of our knowledge, inversion for the geothermal heat flux from surface velocity observations has not been addressed previously.

When formulating the thermomechanically coupled inverse problem, we must assume an appropriate thermal regime. Ice sheets and glaciers can be in one of the following four thermal states: (1) all of the ice is below the melting point; (2) the melting point is reached only at the bed; (3) a basal layer of finite thickness is at melting point; or (4) all of the ice is at the melting point except for a surface layer (Paterson, 1994, p. 205). While in the first case, the thermal basal boundary condition is simply characterized by the geothermal heat flux, in the other three cases, this condition must be modified to include the heat generated by friction at the base and to account for melting. Due to the unknown basal state of the ice, these latter three cases lead to more complex inverse problems, which typically involve variational inequalities in the forward problem. To make the inverse problem tractable, here we assume that all of the ice is below the melting point. We also assume the ice flow is in a steady state.

The inverse problem is formulated as a regularized nonlinear least squares minimization problem governed by thermomechanically coupled nonlinear Stokes and thermal advection-diffusion equations. The cost functional we minimize represents the sum of the squared differences between observed and predicted surface velocities and a regularization term that renders this ill-posed inverse problem well-posed. Discretizing the infinite-dimensional geothermal heat flux field leads to a large-scale PDE-constrained numerical optimization problem; as such, derivative-based optimization methods offer the only hope for its efficient solution (Gunzburger, 2003; Hinze et al., 2009; Borzì and Schulz, 2012; De los Reyes, 2015). In Petra et al. (2012), we presented an infinite-dimensional adjoint-based inexact Gauss-Newton method for the inference of basal friction and rheology



parameters from surface velocity observations and a nonlinear Stokes model of ice sheet flow. Here, we extend our previous work to the present inverse problem of inferring the geothermal heat flux in a thermomechanically coupled ice flow model. This problem also serves as a prototype for a broader class of multiphysics inverse problems.

We systematically study how well finite-amplitude variations of the geothermal heat flux can be recovered from noisy surface velocity observations. To be precise, we invert for geothermal heat flux fields that contain large and small wavelength variations using velocity observations with various degrees of error. Our results show that the quality of the reconstructed geothermal heat flux deteriorates with smaller wavelength variations and with increasing noise level in the observations. In addition, we study the influence of the number of observations and find that the reconstruction improves as the number of observation points increases, provided the discretization of the model equations is sufficiently fine to capture the additional information from a larger number of observations. To analyze prospects and limitations of the inversion, we also investigate the spectrum of the Hessian of the data misfit part of the cost functional, which provides information about directions in parameter space that can be recovered from observations.

A common approach to the numerical solution of multiphysics problems uses operator splitting; namely, motivated by the difficulty of either solving a two-way coupled system, or else by the difficulty of computing the Jacobian of a coupling term, one discards certain coupling terms in the Jacobian of the forward problem to reduce the two-way coupled problem to one that is coupled in one direction. The coupled problem is then solved by iterating back-and-forth between the solution of single physics components. This approach, which we term "one-way coupled," can often yield convergence to the solution of the fully coupled multiphysics problem, depending on the spectral radius of a certain iteration matrix. The one-way coupled approach has been used successfully for the solution of thermomechanically coupled ice sheet forward problems in Dahl-Jensen (1989); Hvidberg (1996); Price et al. (2007); Zwinger et al. (2007); Zhang et al. (2011).

However, when solving the corresponding multiphysics inverse problem using gradient-based methods, the use of such a one-way coupled approach may be problematic. In particular, sacrificing coupling terms in the Jacobian (while often acceptable for the forward problem) will lead to an incorrect adjoint operator, since this operator is given by the transpose of the Jacobian. This approximate adjoint operator leads to an incorrect adjoint solution, which then leads to an incorrect gradient. Since the necessary optimality condition for the inverse problem states that the gradient must vanish, an incorrect gradient leads to the wrong solution of the inverse problem. Moreover, since line search methods require descent on the cost functional in a direction based on the gradient, the inconsistency between the cost functional and its gradient can lead to failure of the line search and thus lack of convergence. Thus, sacrificing coupling terms as is commonly done for the forward problem may not lead to convergent inverse iterations, and if the inverse iterations do converge, they will converge to the wrong inverse solution.

In general, how much of a difference this will make to the solution of the inverse problem will depend on the strength of the coupling terms that have been neglected in the adjoint problem. In particular, despite a gradient that has been computed from an incorrect adjoint equation, and early termination of optimization iterations, one might still obtain a reasonable approximation of the correct inverse solution. To illustrate these issues in the context of a thermomechanically coupled ice sheet inverse problem, we neglect certain coupling terms in the Jacobian (as might be done in a forward solver), leading to an incorrect adjoint operator. We then compare inversion results obtained using an approximate gradient based on a one-way coupled adjoint operator—to




which we refer as a "one-way coupled gradient"—with inversions that use the correct gradient (i.e., based on the fully-coupled adjoint). The results indicate that using this one-way coupled gradient instead of the correct gradient leads to a deterioration in the convergence rate of the inverse solver and eventual failure of the line search, but that the resulting inverse solution for the geothermal flux does not differ substantially from the correct inverse solution.

The remaining sections of this paper are organized as follows. In Section 2, we describe the forward ice sheet problem and the corresponding inverse problem for the geothermal heat flux. In Section 3, we give expressions for the adjoint-based gradient and action of the Hessian of the cost functional. Then, in Section 4 we present the discretization of the forward problem, which involves a stabilization technique applied to prevent oscillatory solutions when the heat equation is advection-dominated, and discuss the optimize-then-discretize and discretize-then-optimize approaches for computing the gradient of the cost functional.

In Section 5, we present inversion results for two- and three-dimensional model problems and in Section 6 we discuss the fully coupled versus one-way coupled approaches to computing the gradient for thermomechanically coupled ice sheet inverse problem. Finally, Section 7 provides concluding remarks.

## 2   Formulation of the inverse problem

In this section, we state the multiphysics forward problem describing the thermomechanically coupled dynamics of ice flow

and formulate an inverse problem, in which we seek to infer the unknown geothermal heat flux at the base of a moving mass of ice from pointwise velocity observations at its top surface. The inverse problem is formulated as a minimization problem with a least squares data misfit cost functional.

### 2.1   The forward problem

Ice sheets and glaciers can be modeled as viscous, incompressible, non-Newtonian, heat-conducting fluids. Assuming the mass

of ice occupying a domain $\Omega$ is in steady state, the balance of mass, linear momentum, and energy state that (Hutter, 1983)

$$\nabla \cdot \boldsymbol{u} = 0, \tag{1}$$

$$-\nabla \cdot \boldsymbol{\sigma_u} = \rho \boldsymbol{g}, \tag{2}$$

$$\rho c \boldsymbol{u} \cdot \nabla \theta - \nabla \cdot (K \nabla \theta) = 2\eta \, \mathrm{tr}(\dot{\boldsymbol{\varepsilon}}_{\boldsymbol{u}}^2), \tag{3}$$

where $\boldsymbol{u}$ is the velocity field, $\theta$ the temperature field, $\boldsymbol{\sigma_u}$ the stress tensor, $\rho$ the density, $\boldsymbol{g}$ the acceleration of gravity, $c$ the

specific heat capacity, and $K$ the thermal conductivity. The stress, $\boldsymbol{\sigma_u}$, can be decomposed as $\boldsymbol{\sigma_u} = \boldsymbol{\tau_u} - \boldsymbol{I}p$, where $\boldsymbol{\tau_u}$ is the deviatoric stress tensor, $p$ the pressure, and $\boldsymbol{I}$ the second-order unit tensor. A commonly employed isotropic constitutive law is Glen's flow law (Glen, 1955),

$$\boldsymbol{\tau_u} = 2\eta(\boldsymbol{u}, \theta)\dot{\boldsymbol{\varepsilon}}_{\boldsymbol{u}}, \ \text{ with } \ \eta(\boldsymbol{u}, \theta) := \frac{1}{2} A(\theta)^{-\frac{1}{n}} \dot{\varepsilon}_{\mathrm{II}}^{\frac{1-n}{2n}}, \tag{4}$$

where $\eta(\boldsymbol{u}, \theta)$ is the effective viscosity, $\dot{\boldsymbol{\varepsilon}}_{\boldsymbol{u}} := \frac{1}{2}(\nabla \boldsymbol{u} + \nabla \boldsymbol{u}^T)$ the strain rate tensor, $\dot{\varepsilon}_{\mathrm{II}} := \frac{1}{2}\mathrm{tr}(\dot{\boldsymbol{\varepsilon}}_{\boldsymbol{u}}^2)$ the second invariant of the

strain rate tensor, and $n$ Glen's flow law exponent. Here, $A$ depends on the ice temperature according to the Arrhenius relation





$A(\theta) = A_0 \exp\left(-\frac{Q}{R\theta}\right)$, where $Q$ is the activation energy, $R$ Boltzmann's constant, and $A_0$ a pre-exponential constant (Paterson, 1994). The appropriate value of Glen's flow law exponent $n$ has been a matter of debate; one could invert for it as a spatial field from surface velocities (Petra et al., 2012). Instead, here we use the constant value $n = 3$, which is typically used in glaciology (Paterson, 1994; Van der Veen, 2013). To avoid singularities in Glen's flow law, we add a small positive number $\epsilon$ to $\dot{\varepsilon}_{II}$ in (4), such that the modified viscosity

$$\eta(\boldsymbol{u}, \theta) := \frac{1}{2} A(\theta)^{-\frac{1}{n}} \left(\dot{\varepsilon}_{II} + \epsilon\right)^{\frac{1-n}{2n}} \tag{5}$$

is bounded from below (Hutter, 1983; Jouvet and Rappaz, 2012).

The energy equation (3) is an advection-diffusion equation for the temperature field with a strain heating term on the right hand side. Note that the Stokes system (1, 2) and the energy equation (3) are two-way coupled: the velocity governed by the Stokes equations is the advection velocity in the energy equation and it additionally enters through the strain heating term on the right side of (3). In the opposite direction, the temperature enters in the Stokes equations through the viscosity term given in (4) and thus affects the flow field.

The domain $\Omega$ is taken as a two- or three-dimensional ice slab with the following boundary conditions. On the top surface, $\Gamma_t$, we impose a traction-free boundary condition for the momentum equation and an imposed temperature for the energy equation. At the base of the ice sheet, $\Gamma_b$, we assume that the ice is below the pressure melting point and frozen to the bedrock. Hence, the boundary conditions are no-slip conditions for the momentum equation and thermal flux conditions for the energy equation representing the flux of geothermal heat into the ice from below (Greve and Blatter, 2009). Additional conditions for the lateral boundaries for the model problems used in our study are specified in Section 5.

In summary, the *forward problem* is given by

$$\nabla \cdot \boldsymbol{u} = 0 \qquad \text{in } \Omega, \tag{6}$$

$$-\nabla \cdot \boldsymbol{\sigma}_{\boldsymbol{u}} = \rho \boldsymbol{g} \qquad \text{in } \Omega, \tag{7}$$

$$\rho c \boldsymbol{u} \cdot \nabla \theta - \nabla \cdot (K \nabla \theta) = 2\eta \operatorname{tr}(\dot{\varepsilon}_{\boldsymbol{u}}^2) \qquad \text{in } \Omega, \tag{8}$$

$$\boldsymbol{\sigma}_{\boldsymbol{u}} \boldsymbol{n} = \boldsymbol{0}, \ \theta = \theta_s \qquad \text{on } \Gamma_t, \tag{9}$$

$$\boldsymbol{u} = \boldsymbol{0}, \ K \nabla \theta \cdot \boldsymbol{n} = G \qquad \text{on } \Gamma_b, \tag{10}$$

+ additional lateral B.C.s,

where $\boldsymbol{n}$ is the outward unit normal vector on $\Gamma_t$ or $\Gamma_b$, $\theta_s$ is the prescribed temperature at the top surface, and $G$ is the geothermal heat flux, and the expressions for the stress $\boldsymbol{\sigma}_{\boldsymbol{u}}$ have been given previously.

Next, we present a weak form of the forward problem (6)–(10), which serves as the basis for the finite element discretization of these equations, and is also used in the Lagrangian functional in Section 3. This weak form is found by multiplying the Stokes system (1, 2) and the energy equation (3) by test functions, integrating over $\Omega$, integrating by parts where appropriate and adding up the three weak equations. The weak form of the forward problem (6)–(10) is thus: Find $(\boldsymbol{u}, p, \theta) \in \mathcal{U} \times \mathcal{P} \times \mathcal{T}$

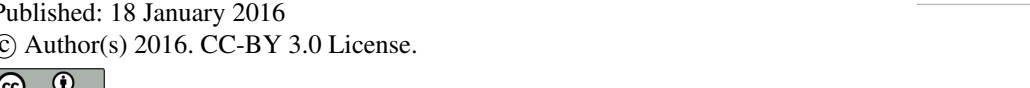



such that

$$a(\boldsymbol{u}, p, \theta; \hat{\boldsymbol{v}}, \hat{q}, \hat{\lambda}) = \langle \rho \boldsymbol{g}, \hat{\boldsymbol{v}} \rangle_\Omega + \langle G, \hat{\lambda} \rangle_{\Gamma_b}, \tag{11}$$

for all test functions $(\hat{\boldsymbol{v}}, \hat{q}, \hat{\lambda}) \in \mathcal{U} \times \mathcal{P} \times \mathcal{T}_0$, where

$$a(\boldsymbol{u}, p, \theta; \hat{\boldsymbol{v}}, \hat{q}, \hat{\lambda}) = \int_\Omega (2\eta(\boldsymbol{u}, \theta)\dot{\boldsymbol{\varepsilon}}_{\boldsymbol{u}} : \dot{\boldsymbol{\varepsilon}}_{\hat{\boldsymbol{v}}} - p\nabla \cdot \hat{\boldsymbol{v}} - \hat{q}\nabla \cdot \boldsymbol{u})\, d\boldsymbol{x} + \int_\Omega (\rho c \hat{\lambda} \boldsymbol{u} \cdot \nabla \theta + K \nabla \theta \cdot \nabla \hat{\lambda})\, d\boldsymbol{x} - \int_\Omega (2\hat{\lambda}\eta(\boldsymbol{u}, \theta)\dot{\boldsymbol{\varepsilon}}_{\boldsymbol{u}} : \dot{\boldsymbol{\varepsilon}}_{\boldsymbol{u}})\, d\boldsymbol{x},$$

and

$$\langle \rho \boldsymbol{g}, \hat{\boldsymbol{v}} \rangle_\Omega = \int_\Omega \rho \boldsymbol{g} \cdot \hat{\boldsymbol{v}}\, d\boldsymbol{x}, \text{ and } \langle G, \hat{\lambda} \rangle_{\Gamma_b} = \int_{\Gamma_b} \hat{\lambda} G\, d\boldsymbol{s}.$$

Here, $\dot{\boldsymbol{\varepsilon}}_{\hat{\boldsymbol{v}}}$ is the strain rate tensor based on $\hat{\boldsymbol{v}}$, and ':' denotes the scalar product between second-order tensors. The spaces in the above equations are defined as

$$\begin{aligned}
\mathcal{U} &= \{\boldsymbol{u} : \Omega \to \mathbb{R}^d \mid \boldsymbol{u}|_{\Gamma_b} = \boldsymbol{0}\}, \\
\mathcal{P} &= \{p : \Omega \to \mathbb{R}\}, \\
\mathcal{T} &= \{\theta : \Omega \to \mathbb{R} \mid \theta|_{\Gamma_t} = \theta_s\}, \\
\mathcal{T}_0 &= \{\hat{\lambda} : \Omega \to \mathbb{R} \mid \hat{\lambda}|_{\Gamma_t} = 0\}, \\
\mathcal{Q} &= \{G : \Gamma_b \to \mathbb{R}\},
\end{aligned} \tag{12}$$

where all functions are assumed to be sufficiently regular for the weak form (11) to be well defined. In the next section, we formulate an inverse problem to infer the unknown geothermal heat flux $G$ present in the basal boundary conditions from surface velocity observations.

## 2.2 The inverse problem

The geothermal heat flux field $G(\boldsymbol{x})$ is, in general, not directly observable and thus uncertain. For instance, the current estimates for $G(\boldsymbol{x})$ in Antarctica differ significantly (Shapiro and Ritzwoller, 2004; Maule et al., 2005). Therefore, our goal is to infer this field from available surface ice velocity observations by exploiting the temperature dependence of the flow, which enters through the dependence of the viscosity on the ice temperature. The inverse problem is formulated as follows: Given (possibly noisy) pointwise observations of the ice surface velocity, $\boldsymbol{u}^{\text{obs}}$, we wish to infer the geothermal heat flux field $G(\boldsymbol{x})$ at the base of the ice sheet that best reproduces the observed velocity via the coupled thermomechanics ice flow model (6)–(10). This can be formulated as the following nonlinear least squares optimization problem,

$$\min_{G \in \mathcal{Q}} \ \mathcal{J}(G) := \frac{1}{2}\|\mathcal{B}\boldsymbol{u}(G) - \boldsymbol{u}^{\text{obs}}\|^2 + \mathcal{R}(G), \tag{13}$$

where the dependence of the velocity $\boldsymbol{u}$ on the geothermal heat flux $G$ is given by the solution of the coupled thermomechanics ice flow model (6)–(10), and $\mathcal{B}$ is an observation operator that maps the surface velocity field to velocity observations at a set of observation points on $\Gamma_t$.





The first term in the cost functional $\mathcal{J}(G)$ is the data misfit that represents the error between the observed velocity field $\boldsymbol{u}^{\text{obs}}$ and that predicted by the thermomechanics ice flow model, $\boldsymbol{u}$. The regularization term $\mathcal{R}(G)$ imposes regularity on the inversion field, such as smoothness. Often, this reflects prior knowledge on the model parameters. In the absence of regularization, the inverse problem is ill-posed; in particular, the solution is not unique in that many model parameter fields may be consistent with the data to within the observational noise, and thus the solution is highly sensitive to errors in the observations (Engl et al., 1996; Vogel, 2002). For instance, as will be discussed in Section 5, small wavelength components in the geothermal heat flux cannot be identified from surface observations, and thus have to be constrained by the regularization. Here we apply a gradient-type Tikhonov regularization, i.e.,

$$\mathcal{R}(G) := \frac{\gamma}{2} \int_{\Gamma_b} |\boldsymbol{T}\nabla G|^2 \, ds, \tag{14}$$

where $\boldsymbol{T} := \boldsymbol{I} - \boldsymbol{n} \otimes \boldsymbol{n}$ is the tangential operator, "$\otimes$" represents the tensor (or outer) product, and $\boldsymbol{I}$ is the second-order unit tensor. This regularization imposes a greater penalty on more oscillatory components of $G$, and, thus, smoothly varying fields are preferred in the inversion of the geothermal heat flux. The regularization parameter $\gamma > 0$ controls the strength of the imposed smoothness relative to the data misfit.

## 3 Solution of the inverse problem via an adjoint-based inexact Newton method

To compute the minimizer for the large-scale optimization problem (13), we employ a derivative-based descent method and thus require derivatives of the nonlinear least squares optimization problem (13) with respect to the parameter $G$. To improve over linearly-convergent methods (such as nonlinear conjugate gradients) or superlinearly-convergent methods (such as limited memory BFGS), here we advocate a Newton method, which employs Hessian information (i.e., second derivatives) to provide an (asymptotic) quadratic convergence rate. Moreover, typically Newton's method converges in a number of iterations that is independent of the parameter dimension/mesh size, which is typically not true of gradient-only methods (Petra et al., 2012). Beyond the faster convergence, in our experience the use of Hessian information for highly nonlinear inverse problems such as those involving ice sheet models is typically able to obtain another one or two orders of magnitude reduction in the norm of the gradient, leading to the extraction of additional details in the reconstructed parameter field.

Starting with an initial guess for the parameter field $G$, Newton's method iteratively updates the parameter field based on minimizing a sequence of quadratic approximations of the cost functional, $\mathcal{J}$, using gradient and Hessian information of $\mathcal{J}$ with respect to $G$. That is, the parameter is updated by

$$G_{\text{new}} = G + \alpha \tilde{G}, \tag{15}$$

where $G$ is the current model parameter field, $\alpha$ is the step length, appropriately chosen so that the cost functional $\mathcal{J}$ is sufficiently decreased at each iteration, and $\tilde{G}$ is the direction which is obtained by solving the linear system

$$\mathcal{H}(G)(\tilde{G}) = -\mathcal{G}(G). \tag{16}$$



Here, $\mathcal{G}(G)$ and $\mathcal{H}(G)$ denote the gradient and the Hessian of the least squares cost functional $\mathcal{J}$, respectively, evaluated at the current parameter field $G$.

In this section, we provide expressions for the gradient $\mathcal{G}(G)$ and Hessian $\mathcal{H}(G)$. For the efficient computation of gradient and Hessian operators, we employ adjoint methods (see, for example, (Gunzburger, 2003; Tröltzsch, 2010; Borzì and Schulz, 2012)). All expressions in this section are given in infinite-dimensional form, which has several advantages compared to discretizing the optimization problem first and then differentiating. First, one avoids differentiating through artifacts of the discretization or solver, which may not even be differentiable. Second, it is much easier and "cleaner" to derive gradient and Hessian information at the infinite-dimensional level. Third, the resulting expressions are in weak form, which provides a natural and systematic path to discretization by Galerkin finite elements. The downside to differentiating at the infinite-dimensional level is that the resulting gradient and Hessian expressions may not be "consistent." These issues will be discussed in the next section.

In what follows, we use the formal Lagrange approach, which computes the gradient by taking variations of a Lagrangian functional. The Lagrangian functional $\mathcal{L}$ combines the cost functional (13) with the weak form (11) of the forward problem, with test functions $\hat{\boldsymbol{v}}, \hat{q}$ and $\hat{\lambda}$ becoming the adjoint velocity $\boldsymbol{v}$, adjoint pressure $q$, and adjoint temperature $\lambda$:

$$\mathcal{L}(\boldsymbol{u}, p, \theta; \ \boldsymbol{v}, q, \lambda; \ G) := \mathcal{J}(G) + a(\boldsymbol{u}, p, \theta; \boldsymbol{v}, q, \lambda) - \langle \rho \boldsymbol{g}, \boldsymbol{v} \rangle_\Omega - \langle G, \lambda \rangle_{\Gamma_b}. \tag{17}$$

The gradient of $\mathcal{J}$ with respect to the unknown heat flux $G$ is found by requiring that variations of the Lagrangian $\mathcal{L}$ with respect to the forward and adjoint variables vanish. The gradient $\mathcal{G}(G)$ is then found by taking the variation of $\mathcal{L}$ with respect to $G$. In strong form, the gradient evaluated at $G$, $\mathcal{G}(G)$, is then given by:

$$\mathcal{G}(G) := \begin{cases} -\nabla \cdot (\gamma \boldsymbol{T} \nabla G) - \lambda & \text{on } \Gamma_b, \\ (\gamma \boldsymbol{T} \nabla G) \cdot \overline{\boldsymbol{n}} & \text{on } \partial\Gamma_b, \end{cases} \tag{18}$$

where $\overline{\boldsymbol{n}}$ is the outer normal vector on $\partial\Gamma_b$. Variations of $\mathcal{L}$ with respect to the adjoint variables $(\boldsymbol{v}, q, \lambda)$ simply recover the forward problem. On the other hand, variations of $\mathcal{L}$ with respect to the forward variables $(\boldsymbol{u}, p, \theta)$ yield the so-called *adjoint problem*, which is given in strong form by

$$\nabla \cdot \boldsymbol{v} = 0 \qquad\qquad \text{in } \Omega, \tag{19}$$

$$-\nabla \cdot \boldsymbol{\sigma_v} = -\rho c \lambda \nabla \theta \qquad\qquad \text{in } \Omega, \tag{20}$$

$$-\rho c \boldsymbol{u} \cdot \nabla \lambda - \nabla \cdot (K \nabla \lambda) = F_\lambda \qquad\qquad \text{in } \Omega, \tag{21}$$

$$\lambda = 0, \ \boldsymbol{\sigma_v} \boldsymbol{n} = \mathcal{B}^*(\boldsymbol{u}^{\text{obs}} - \mathcal{B}\boldsymbol{u}) \qquad\qquad \text{on } \Gamma_t, \tag{22}$$

$$\boldsymbol{v} = \boldsymbol{0}, \ K \nabla \lambda \cdot \boldsymbol{n} = 0 \qquad\qquad \text{on } \Gamma_b, \tag{23}$$

+ additional lateral B.C.s.

The adjoint stress $\boldsymbol{\sigma_v}$ in (20) depends on the forward velocity $\boldsymbol{u}$ and the temperature $\theta$, and is given by

$$\boldsymbol{\sigma_v} := 2\eta(\boldsymbol{u}, \theta) \left[ \left( \mathsf{I} + \frac{1-n}{2n} \frac{\dot{\boldsymbol{\varepsilon}}_{\boldsymbol{u}} \otimes \dot{\boldsymbol{\varepsilon}}_{\boldsymbol{u}}}{\dot{\varepsilon}_{\text{II}} + \epsilon} \right) \dot{\boldsymbol{\varepsilon}}_{\boldsymbol{v}} - \frac{1+n}{n} \lambda \dot{\boldsymbol{\varepsilon}}_{\boldsymbol{u}} \right] - \boldsymbol{I}q,$$





where $\dot{\boldsymbol{\varepsilon}}_{\boldsymbol{v}}$ is the adjoint strain rate tensor and is given by $\frac{1}{2}(\nabla\boldsymbol{v}+\nabla\boldsymbol{v}^T)$, $\mathsf{I}$ is the fourth-order identity tensor, and "$\otimes$" represents the tensor (or outer) product between second-order tensors. The right hand side in the adjoint energy equation (21) is given by

$$F_\lambda := -\frac{2Q\eta}{nR\theta^2}(\lambda\dot{\boldsymbol{\varepsilon}}_{\boldsymbol{u}}:\dot{\boldsymbol{\varepsilon}}_{\boldsymbol{u}} - \dot{\boldsymbol{\varepsilon}}_{\boldsymbol{u}}:\dot{\boldsymbol{\varepsilon}}_{\boldsymbol{v}}).$$

As can be seen from (19)–(23), the adjoint problem is driven by the misfit between observed and predicted surface velocity on the top boundary, i.e., $\mathcal{B}^*(\boldsymbol{u}^{\text{obs}} - \mathcal{B}\boldsymbol{u})$. Since the observations are of ice velocity on the top surface $\Gamma_t$, the data misfit shows up in the adjoint problem as a source term for the Neumann boundary condition on $\Gamma_t$, (22). Observations in the interior of $\Omega$ would amount to a similar contribution on the right hand side of (20). Since the adjoint equation depends on $(\boldsymbol{u}, p, \theta)$, each gradient computation also requires the solution of the forward problem (6)–(10). Solution of the adjoint problem (19)–(23) provides the adjoint temperature $\lambda$ needed to evaluate the gradient in (18).

Now that the computation of the gradient, which forms the right-hand side of the Newton system (16) has been described, we present the computation of the Hessian operator, $\mathcal{H}$, on the left-hand side of the Newton system. We note that explicitly forming and storing the Hessian matrix resulting upon discretization is not an option, since computing each column would require at least a linearized forward solve. Instead, we solve the Newton system (16) using the linear conjugate gradient (CG) method, which does not require the explicit Hessian, but only the action of the Hessian on a vector at each CG iteration. We next present expressions for this Hessian action on vectors in terms of the solution of a pair of linearized forward and adjoint problems. These expressions are simply stated here; an analogous derivation, for the isothermal case, is presented in (Petra et al., 2012). The action of the Hessian operator in a given CG direction $\tilde{G}$, evaluated at the current iterate, $G$, can be expressed in strong form as

$$\mathcal{H}(G)(\tilde{G}) := \begin{cases} -\nabla \cdot (\gamma \boldsymbol{T} \nabla\tilde{G}) - \tilde{\lambda} & \text{on } \Gamma_b, \\ (\gamma \boldsymbol{T} \nabla\tilde{G}) \cdot \overline{\boldsymbol{n}} & \text{on } \partial\Gamma_b. \end{cases} \tag{24}$$

Beyond the forward and adjoint equations that must be solved to evaluate the gradient, the Hessian action requires two additional forward-like equations: the *incremental forward* and *incremental adjoint* equations. These can be derived by constructing a new Lagrangian for the gradient, which imposes both the forward and adjoint equations as constraints, and taking variations with respect to the Lagrange multipliers that enforce these equations. The resulting *incremental forward problem*, which is to be solved for the incremental forward velocity, pressure, and temperature variables, $(\tilde{\boldsymbol{u}}, \tilde{p}, \tilde{\theta})$, is given by

$$\nabla \cdot \tilde{\boldsymbol{u}} = 0 \qquad\qquad \text{in } \Omega, \tag{25}$$

$$-\nabla \cdot \boldsymbol{\sigma}_{\tilde{\boldsymbol{u}}} = \boldsymbol{0} \qquad\qquad \text{in } \Omega, \tag{26}$$

$$\rho c \boldsymbol{u} \cdot \nabla\tilde{\theta} - \nabla \cdot (K\nabla\tilde{\theta}) = F_{\tilde{\theta}} \qquad\qquad \text{in } \Omega, \tag{27}$$

$$\tilde{\theta} = 0, \ \boldsymbol{\sigma}_{\tilde{\boldsymbol{u}}}\boldsymbol{n} = \boldsymbol{0} \qquad\qquad \text{on } \Gamma_t, \tag{28}$$

$$\tilde{\boldsymbol{u}} = \boldsymbol{0}, \ K\nabla\tilde{\theta} \cdot \boldsymbol{n} = \tilde{G} \qquad\qquad \text{on } \Gamma_b, \tag{29}$$

$+$ additional lateral B.C.s.





with

$$\boldsymbol{\sigma}_{\tilde{\boldsymbol{u}}} := 2\eta(\boldsymbol{u},\theta)\left[\left(\mathsf{I} + \frac{1-n}{2n}\frac{\dot{\boldsymbol{\varepsilon}}_{\boldsymbol{u}} \otimes \dot{\boldsymbol{\varepsilon}}_{\boldsymbol{u}}}{\dot{\varepsilon}_{\mathrm{II}}+\epsilon}\right)\dot{\boldsymbol{\varepsilon}}_{\tilde{\boldsymbol{u}}} - \frac{Q}{nR\theta^2}\tilde{\theta}\dot{\boldsymbol{\varepsilon}}_{\boldsymbol{u}}\right] - \boldsymbol{I}\tilde{p},$$

$$F_{\tilde{\theta}} := -\rho c\tilde{\boldsymbol{u}}\cdot\nabla\theta + 2\eta\left(\frac{1+n}{n}\dot{\boldsymbol{\varepsilon}}_{\boldsymbol{u}}:\dot{\boldsymbol{\varepsilon}}_{\tilde{\boldsymbol{u}}} - \frac{Q}{nR\theta^2}\tilde{\theta}\dot{\boldsymbol{\varepsilon}}_{\boldsymbol{u}}:\dot{\boldsymbol{\varepsilon}}_{\boldsymbol{u}}\right).$$

5 Note that the incremental forward problem (25)–(29) resembles the forward problem, and in fact corresponds to a linearized (with respect to all variables) version of it. Both the operator and the right hand side depend on the forward variables $(\boldsymbol{u},p,\theta)$, and the right hand side also depends on the CG direction $\tilde{G}$.

The resulting *incremental adjoint problem*, to be solved for the incremental adjoint velocity, pressure, and temperature $(\tilde{\boldsymbol{v}},\tilde{q},\tilde{\lambda})$, is then given by

$$\nabla\cdot\tilde{\boldsymbol{v}} = 0 \qquad\qquad \text{in } \Omega, \tag{30}$$

$$-\nabla\cdot\boldsymbol{\sigma}_{\tilde{\boldsymbol{v}}} + \rho c\tilde{\lambda}\nabla\theta = \nabla\cdot\boldsymbol{\tau}_{\boldsymbol{v}} - \rho c\lambda\nabla\tilde{\theta} \qquad\qquad \text{in } \Omega, \tag{31}$$

$$-\rho c\boldsymbol{u}\cdot\nabla\tilde{\lambda} - \nabla\cdot(K\nabla\tilde{\lambda}) = F_{\tilde{\lambda}} + \hat{F} \qquad\qquad \text{in } \Omega, \tag{32}$$

$$\tilde{\lambda} = 0, \ \boldsymbol{\sigma}_{\tilde{\boldsymbol{v}}}\boldsymbol{n} = -\mathcal{B}^*\mathcal{B}\tilde{\boldsymbol{u}} - \boldsymbol{\tau}_{\boldsymbol{v}}\boldsymbol{n} \qquad\qquad \text{on } \Gamma_t, \tag{33}$$

$$\tilde{\boldsymbol{v}} = \boldsymbol{0}, \ K\nabla\tilde{\lambda}\cdot\boldsymbol{n} = 0 \qquad\qquad \text{on } \Gamma_b, \tag{34}$$

15 + additional lateral B.C.s.

with

$$\boldsymbol{\sigma}_{\tilde{\boldsymbol{v}}} := 2\eta(\boldsymbol{u},\theta)\left[\left(\mathsf{I} + \frac{1-n}{2n}\frac{\dot{\boldsymbol{\varepsilon}}_{\boldsymbol{u}} \otimes \dot{\boldsymbol{\varepsilon}}_{\boldsymbol{u}}}{\dot{\varepsilon}_{\mathrm{II}}+\epsilon}\right)\dot{\boldsymbol{\varepsilon}}_{\tilde{\boldsymbol{v}}} - \frac{1+n}{n}\tilde{\lambda}\dot{\boldsymbol{\varepsilon}}_{\boldsymbol{u}}\right] - \boldsymbol{I}\tilde{q},$$

$$\begin{aligned}\boldsymbol{\tau}_{\boldsymbol{v}} \ := \ & 2\eta(\boldsymbol{u},\theta)\left[\frac{1-n}{2n}\left(\frac{\dot{\boldsymbol{\varepsilon}}_{\boldsymbol{u}}:\dot{\boldsymbol{\varepsilon}}_{\boldsymbol{v}}}{\dot{\varepsilon}_{\mathrm{II}}+\epsilon}\mathsf{I} + \frac{\dot{\boldsymbol{\varepsilon}}_{\boldsymbol{u}} \otimes \dot{\boldsymbol{\varepsilon}}_{\boldsymbol{v}} + \dot{\boldsymbol{\varepsilon}}_{\boldsymbol{v}} \otimes \dot{\boldsymbol{\varepsilon}}_{\boldsymbol{u}}}{\dot{\varepsilon}_{\mathrm{II}}+\epsilon}\right)\dot{\boldsymbol{\varepsilon}}_{\tilde{\boldsymbol{u}}} + \frac{1-n}{2n}\frac{1-3n}{2n}\frac{(\dot{\boldsymbol{\varepsilon}}_{\boldsymbol{u}}:\dot{\boldsymbol{\varepsilon}}_{\boldsymbol{v}})(\dot{\boldsymbol{\varepsilon}}_{\boldsymbol{u}} \otimes \dot{\boldsymbol{\varepsilon}}_{\boldsymbol{u}})}{(\dot{\varepsilon}_{\mathrm{II}}+\epsilon)^2}\dot{\boldsymbol{\varepsilon}}_{\tilde{\boldsymbol{u}}} \\ & -\frac{1+n}{n}\lambda\left(\mathsf{I} + \frac{1-n}{2n}\frac{\dot{\boldsymbol{\varepsilon}}_{\boldsymbol{u}} \otimes \dot{\boldsymbol{\varepsilon}}_{\boldsymbol{u}}}{\dot{\varepsilon}_{\mathrm{II}}+\epsilon}\right)\dot{\boldsymbol{\varepsilon}}_{\tilde{\boldsymbol{u}}} + \frac{Q}{nR\theta^2}\tilde{\theta}\left(\frac{1+n}{n}\lambda\dot{\boldsymbol{\varepsilon}}_{\boldsymbol{u}} - \left(\mathsf{I} + \frac{1-n}{2n}\frac{\dot{\boldsymbol{\varepsilon}}_{\boldsymbol{u}} \otimes \dot{\boldsymbol{\varepsilon}}_{\boldsymbol{u}}}{\dot{\varepsilon}_{\mathrm{II}}+\epsilon}\right)\dot{\boldsymbol{\varepsilon}}_{\boldsymbol{v}}\right)\right],\end{aligned}$$

$$F_{\tilde{\lambda}} := \frac{Q}{nR\theta^2}\eta(\boldsymbol{u},\theta)\left(\dot{\boldsymbol{\varepsilon}}_{\boldsymbol{u}}:\dot{\boldsymbol{\varepsilon}}_{\tilde{\boldsymbol{v}}} - \tilde{\lambda}\dot{\boldsymbol{\varepsilon}}_{\boldsymbol{u}}:\dot{\boldsymbol{\varepsilon}}_{\boldsymbol{u}}\right),$$

$$\begin{aligned}\hat{F} \ := \ & \rho c\tilde{\boldsymbol{u}}\cdot\nabla\lambda + \frac{Q\eta(\boldsymbol{u},\theta)}{nR\theta^2}\dot{\boldsymbol{\varepsilon}}_{\tilde{\boldsymbol{u}}}:\dot{\boldsymbol{\varepsilon}}_{\boldsymbol{v}} + \frac{Q\eta(\boldsymbol{u},\theta)}{nR\theta^2}\left(\frac{1-n}{2n}\frac{\dot{\boldsymbol{\varepsilon}}_{\boldsymbol{u}}:\dot{\boldsymbol{\varepsilon}}_{\tilde{\boldsymbol{u}}}}{\dot{\varepsilon}_{\mathrm{II}}+\epsilon}\dot{\boldsymbol{\varepsilon}}_{\boldsymbol{u}}:\dot{\boldsymbol{\varepsilon}}_{\boldsymbol{v}} - \frac{1+n}{n}\lambda\dot{\boldsymbol{\varepsilon}}_{\boldsymbol{u}}:\dot{\boldsymbol{\varepsilon}}_{\tilde{\boldsymbol{u}}}\right) \\ & -\frac{2Q\tilde{\theta}\eta(\boldsymbol{u},\theta)}{nR\theta^2}\left(\frac{Q}{nR\theta^2} + \frac{2}{\theta^2}\right)\left(\dot{\boldsymbol{\varepsilon}}_{\boldsymbol{u}}:\dot{\boldsymbol{\varepsilon}}_{\boldsymbol{v}} - \lambda\dot{\boldsymbol{\varepsilon}}_{\boldsymbol{u}}:\dot{\boldsymbol{\varepsilon}}_{\boldsymbol{u}}\right).\end{aligned}$$

25 Note that the incremental adjoint problem (30)–(34) resembles the adjoint problem and is in fact its linearization with respect to the forward and adjoint variables and the unknown geothermal heat flux. Its operator depends on the forward variables only (as does the incremental forward operator), while its right hand side source terms depend on not just the forward variables (as does the incremental forward problem), but on the adjoint and incremental forward variables as well.





In conclusion, to evaluate the expression for the gradient (18) for a given value of the geothermal heat flux $G$, we first solve the forward problem (6)–(10), followed by the adjoint problem (19)–(23) (given the forward solution). To then evaluate the Hessian action (24) in a given direction $\tilde{G}$ at each CG iteration, we solve the incremental forward problem (25)–(29) (given the forward solution), and then solve the incremental adjoint equation (30)–(34) (given the forward, adjoint, and incremental forward solutions).

It is well known that the Newton update direction computed by solving (16) is a descent direction only if the Hessian is positive definite, which is only guaranteed close to a minimizer (Nocedal and Wright, 2006). As in (Petra et al., 2012), the remedy we apply here is to neglect terms in the Hessian expression that involve the adjoint variable, that is, the terms high-lighted in blue in (31)–(33). This leads to the so-called Gauss-Newton approximation of the Hessian, which (with appropriate regularization) is guaranteed to be positive definite. Moreover, since accurate solution of the Newton system (16) is needed only close to the minimum of the regularized data misfit functional $\mathcal{J}$, we terminate the CG iterations early for iterates that are far from the converged solution. This so-called *inexact Newton* method terminates the CG iterations when the norm of the residual of the linear system (16) drops below a tolerance that is proportional to the norm of the gradient, i.e. we terminate at CG iteration $i$ when

$$\|\mathcal{H}(G)(\tilde{G}^i) + \mathcal{G}(G)\| \le \beta \|\mathcal{G}(G)\|,$$

where the so-called forcing term $\beta$ itself can depend on $\|\mathcal{G}(G)\|$. Far from the minimum—when the relative gradient is large— the tolerance is also large, and the CG iterations are terminated early to prevent oversolving. As the minimum is approached, the norm of the gradient decreases, thereby enforcing an increasingly more accurate solution of the Newton system (16). The criterion above is often able to significantly reduce the overall number of CG iterations—and thus the required number of incremental forward/adjoint solves—while still maintaining fast local convergence. When $\beta$ is taken as the order of square root of the gradient, the inexact Newton method retains superlinear convergence (Eisenstat and Walker, 1996).

It is critical that the total number of CG iterations be as small as possible, since as mentioned above, each iteration requires a pair of forward/adjoint incremental problem solves. Despite the significant reduction in overall number of CG iterations provided by inexact solution of the Newton step, the number can still be large if a good preconditioner is not used. An effective preconditioner is simply the inverse of the regularization operator, which amounts to a Laplacian solve on the basal surface. This is because the Hessian of the data misfit operator, like many ill-posed infinite-dimensional inverse operators, has eigenvalues that decay to zero; preconditioning by an inverse Laplacian simply increases the rate of decay. Thus the resulting preconditioned Hessian behaves like a compact perturbation of the identity with smooth dominant eigenfunctions, for which CG converges rapidly and in a number of iterations that is independent of the mesh size; see, for example, (Isaac et al., 2015). This is the preconditioner we use in the numerical examples below.

Once a descent direction is computed by inexact solution of the Newton step equation (16), we must guarantee that sufficient decrease in $\mathcal{J}$ is obtained in that direction so that convergence of the iterations can be assured. This is achieved by a *line search* that finds a step size $\alpha$ satisfying the so-called Armijo condition (Nocedal and Wright, 2006), which has the attractive property that it requires only cost functional evaluations, and not gradient information. The Newton iterations are repeated until the





norm of the gradient of $\mathcal{J}$ is sufficiently small. The inexact Newton method is summarized in Algorithm 1, in which we use $\mu = 0.5$ and $\nu = 10^{-4}$ for the line search.

---

**Algorithm 1** ADJOINT-BASED INEXACT NEWTON

---

Initialize/define variables $G_1, \alpha, \mu, \nu, \varepsilon_{\text{tol}}$

**for** $k = 1, \dots$ **do**

    $(\boldsymbol{u}_k, p_k, \theta_k) \leftarrow$ solve the forward equation with $G_k$

    $(\boldsymbol{v}_k, q_k, \lambda_k) \leftarrow$ solve the adjoint equation with $(\boldsymbol{u}_k, \theta_k)$

    $\mathcal{G}_k \leftarrow$ compute the discrete gradient

    **if** $||\mathcal{G}_k|| < \varepsilon_{\text{tol}}$ **then**

        converged

    **end if**

    Perform preconditioned inexact CG iterations for solving $\mathcal{H}_k \tilde{G}_k = -\mathcal{G}_k$ to compute $\tilde{G}_k$ (each iteration requires solution of a pair of incremental forward/adjoint problems)

    $\alpha \leftarrow 1, \text{descent} = 0$

    **while** descent $= 0$ **do**

        $G_{k+1} \leftarrow G_k + \alpha \tilde{G}_k$

        Solve the forward equation with $G_{k+1}$

        **if** $\mathcal{J}(G_{k+1}) \leq \mathcal{J}(G_k) + \nu \alpha \langle \mathcal{G}_k, \tilde{G}_k \rangle_{\Gamma_b}$ **then**

            descent $= 1$

        **else**

            $\alpha \leftarrow \mu \alpha$

        **end if**

    **end while**

**end for**

---

## 4 Discretization

In this section, we describe the discretization of the forward and the inverse problems and discuss a stabilization technique required to avoid oscillations for advection-dominated problems. We compare two approaches for computating the gradient of the cost functional, namely the optimize-then-discretize (OTD) and the discretize-then-optimize (DTO) approaches.

### 4.1 Discretization of the forward problem and SUPG stabilization

For advection-dominated problems, the standard Galerkin finite element method applied to the energy equation (3) can result in strongly oscillatory solutions, unless the mesh size is less than $2K/(\rho c |\boldsymbol{u}|)$, which results in a smaller critical mesh size as the Peclet number increases. To avoid this onerous mesh size restriction for high Peclet flows, we discretize (3) with the Streamline





Upwind Petrov-Galerkin (SUPG) method (Brooks and Hughes, 1982), which suppresses oscillations on coarser meshes. The SUPG method adds a stabilization term to the standard Galerkin weak form. This term involves the element residual and thus vanishes at the exact solution, so preserving the correct solution of the energy equation in the limit of infinitesimal mesh size.

We use quadratic elements for temperature, and the Taylor-Hood element pair for velocity and pressure (quadratic elements for velocity and linear elements for pressure). We let $\Omega' = \{\Omega_e\}$ be a family of quadrilateral elements of $\Omega$, denoting by $Q_1(\Omega_e)$ bilinear or trilinear functions (in $\mathbb{R}^2$ and $\mathbb{R}^3$, respectively) and by $Q_2(\Omega_e)$ biquadratic or triquadratic functions (in $\mathbb{R}^2$ and $\mathbb{R}^3$, respectively) defined on $\Omega_e$. The discretized spaces are then given by:

$$
\begin{aligned}
\mathcal{U}^h &= \{\boldsymbol{u}^h \in \mathcal{U} : \boldsymbol{u}^h|_{\Omega_e} \in Q_2(\Omega_e)^d \, \forall \, \Omega_e \in \Omega'\}, \\
\mathcal{P}^h &= \{p^h \in \mathcal{P} : P^h|_{\Omega_e} \in Q_1(\Omega_e) \forall \Omega_e \in \Omega'\}, \\
\mathcal{T}^h &= \{\theta^h \in \mathcal{T} : \theta^h|_{\Omega_e} \in Q_2(\Omega_e) \, \forall \, \Omega_e \in \Omega'\}, \\
\mathcal{T}_0^h &= \{\hat{\lambda}^h \in \mathcal{T}_0 : \hat{\lambda}^h|_{\Omega_e} \in Q_2(\Omega_e) \, \forall \, \Omega_e \in \Omega'\}, \\
\mathcal{Q}^h &= \{G^h \in \mathcal{Q} : G^h|_{\Omega_e} \in Q_1(\Omega_e) \, \forall \, \Omega_e \in \Omega'\}.
\end{aligned}
\tag{35}
$$

The SUPG-stabilized discretization of (11) is thus as follows: Find $(\boldsymbol{u}^h, p^h, \theta^h) \in \mathcal{U}^h \times \mathcal{P}^h \times \mathcal{T}^h$ such that

$$
a_s(\boldsymbol{u}^h, p^h, \theta^h; \hat{\boldsymbol{v}}^h, \hat{q}^h, \hat{\lambda}^h) = \langle \rho \boldsymbol{g}, \hat{\boldsymbol{v}}^h \rangle_\Omega + \langle G^h, \hat{\lambda}^h \rangle_{\Gamma_b}
\tag{36}
$$

for all $(\hat{\boldsymbol{v}}^h, \hat{q}^h, \hat{\lambda}^h) \in \mathcal{U}^h \times \mathcal{P}^h \times \mathcal{T}_0^h$, where

$$
a_s(\boldsymbol{u}^h, p^h, \theta^h; \hat{\boldsymbol{v}}^h, \hat{q}^h, \hat{\lambda}^h) = a(\boldsymbol{u}^h, p^h, \theta^h; \hat{\boldsymbol{v}}^h, \hat{q}^h, \hat{\lambda}^h) + \sum_e \langle \tau_e \rho c \boldsymbol{u}^h \cdot \nabla \hat{\lambda}^h, R_\theta \rangle_{\Omega_e},
\tag{37}
$$

with the residual $R_\theta$ of the forward energy equation in (8) given by

$$
R_\theta(\boldsymbol{u}^h, \theta^h) := \rho c \boldsymbol{u}^h \cdot \nabla \theta^h - K \Delta \theta^h - \eta(\boldsymbol{u}^h, \theta^h) \dot{\boldsymbol{\varepsilon}}_{\boldsymbol{u}^h} : \dot{\boldsymbol{\varepsilon}}_{\boldsymbol{u}^h}.
\tag{38}
$$

As can be seen in (37), the test function for the energy equation residual is a multiple of $\boldsymbol{u}^h \cdot \nabla \hat{\lambda}^h$ and, in particular, depends on $\boldsymbol{u}^h$. The stabilization factor $\tau_e \geq 0$ controls the weight of the stabilization term and influences the quality of the discrete solution. Often, by analogy with the optimal one-dimensional choice, $\tau_e = (\coth(Pe) - 1/Pe)h/(2\rho c|\boldsymbol{u}|)$, where $h$ is the element diameter in the direction of the advective velocity $\boldsymbol{u}$, and $Pe = \rho c|\boldsymbol{u}|h/2K$ is the local Peclet number, which determines whether the problem is locally convection-dominated or diffusion dominated (Brooks and Hughes, 1982). The introduction of this stabilization term makes (36) a non-Galerkin discretization, which has consequences for the computation of the derivatives of $\mathcal{J}(G)$ as defined in (13), which is the subject of the next section.

### 4.2 Optimize-then-discretize (OTD) versus discretize-then-optimize (DTO)

The numerical solution of the inverse problem requires the computation of gradients of $\mathcal{J}$ with respect to $G$. These gradients are computed using an adjoint system of equations, and there are two approaches: the optimize-then-discretize (OTD) and the discretize-then-optimize (DTO) approach. In OTD, one derives the adjoint equations at the infinite-dimensional (i.e., the PDE) level, and then discretizes both the forward system (6)-(10) and the adjoint system (19)–(23) independently. As a consequence,





one would then use SUPG stabilization for the forward and adjoint energy equations. In DTO, the forward problem and the cost functional $\mathcal{J}(\cdot)$ are discretized first, resulting in a finite-dimensional optimization problem. Then, for this discretized optimization problem, gradients are computed using a finite-dimensional Lagrangian function, resulting in a finite-dimensional system of adjoint equations. For more details, we refer the reader to Gunzburger (2003) and Hinze et al. (2009).

For Galerkin discretizations, OTD and DTO usually coincide, i.e., they result in exactly the same finite-dimensional gradient. However, the operations of optimization and discretization do not commute when the forward problem is discretized by SUPG. For OTD, the adjoint energy equation (21) is also an advection-diffusion equation, but with advection velocity $-\boldsymbol{u}$. Thus, it also requires stabilization. If SUPG is used to stabilize the adjoint equation, the discrete gradient becomes inconsistent with the discrete cost functional. This is because the discrete adjoint of SUPG stabilization for the forward equation is not equivalent

to SUPG stabilization of the adjoint equation. The implication of an inconsistent gradient is that the computed gradient may not actually lead to a direction of descent with respect to the discretized cost functional, which can result in a failure in the line search and lack of convergence. In the DTO approach, the SUPG stabilization term in (37) produces a contribution in the adjoint equation that has a stabilizing effect. However, this contribution is not a weighted residual of the continuous adjoint energy equation, which can degrade the convergence of the discrete adjoint temperature to the continuous adjoint temperature

(Collis and Heinkenschloss, 2002). On the other hand, the resulting gradient is consistent with the discrete cost functional, and therefore convergence is guaranteed with a Gauss-Newton method and an appropriate line search.

    Both DTO and OTD approaches have advantages and disadvantages, and the preference for one over the other depends on the circumstances of the problem at hand. For sufficiently smooth problems, the differences between OTD and DTO diminish as the mesh size is reduced, and the approaches are equivalent in the limit. In the numerical results of the next section, we

choose DTO so that we can be assured a direction of descent without having to refine the mesh beyond what is necessary for accurate approximation of the forward, adjoint, and parameter fields. That means that the resulting expressions for the discrete gradient and Hessian constructed via the DTO approach are just approximations of the expressions for the continuous gradient and Hessian presented in the previous section, and will converge to those expressions as the mesh is refined. Thus, the infinite-dimensional expressions provided in the previous section continue to provide useful intuition on the nature of the

gradient and Hessian action (for example the resemblance of the incremental forward and adjoint operators to the forward and adjoint operators, the 4th order anisotropy of the effective viscosity in the adjoint and incremental operators, the role of the boundary conditions, etc.).

## 5    Numerical results and discussion

In this section, we study properties of the multiphysics inverse problem to infer the unknown geothermal heat flux field from

30 surface velocity observations. In particular, we study the limits of our ability to invert for the heat flux as a function of the length scales of the heat flux and of the noise level in the velocity observations.

    We consider a two- and a three-dimensional ice slab, $\Omega$, of length $L = 80$ km and thickness $s$ given by

$$s(x) = (H - H_0)\cos\left(\frac{\pi x}{2L}\right) + H_0, \tag{39}$$



where $x \in [0, L]$, $H = 2$ km is the maximum ice thickness, and $H_0 = 0.1$ km is the ice thickness at the outflow boundary $\Gamma_o$. The coordinate system and the ice slab domain for the two-dimensional problem are shown in Figure 1; the three-dimensional geometry is an extrusion in the $y$-direction of this geometry.

In all model problems, we assume that the surface temperature increases as the elevation decreases as follows (see also Figure 3):

$$\theta_s(x) = \theta_0 + a(H - s(x)), \tag{40}$$

where $\theta_0 = -50\ ^\circ\text{C}$ is the temperature at $x = 0$, and $a$ is the lapse rate, taken to be $a = 6.5\ ^\circ\text{Ckm}^{-1}$.

The boundary conditions are as follows:

- on the top surface, $\Gamma_t$, we assume zero traction for the velocity and the surface temperature $\theta_s$ defined in (40), i.e.,

  $$\boldsymbol{\sigma_u n} = \boldsymbol{0}, \ \theta = \theta_s;$$

- on the bottom surface $\Gamma_b$, we assume that the ice is frozen to the bedrock, i.e., we apply a no sliding condition, and assume a geothermal heat flux condition for the temperature, i.e.,

  $$\boldsymbol{u} = \boldsymbol{0}, \ K\nabla\theta \cdot \boldsymbol{n} = G;$$

- on the outflow boundary, $\Gamma_o$, we ignore the atmospheric stress (i.e., the atmospheric pressure and wind stress), which is small compared to the typical stresses in an ice sheet, and thus impose a traction-free condition; the surface temperature $\theta_s$ is as defined in (40), i.e.,

  $$\boldsymbol{\sigma_u n} = \boldsymbol{0}, \ \theta = \theta_s;$$

- we assume that $\Gamma_i$ is an ice divide, i.e., there is no inflow, no shear stress and no heat flux, i.e.,

  $$\boldsymbol{u} \cdot \boldsymbol{n} = 0, \ \boldsymbol{T\sigma_u n} = \boldsymbol{0}, \ K\nabla\theta \cdot \boldsymbol{n} = 0;$$

- in addition, for the three-dimensional problem, we impose periodic boundary conditions on the fore and aft boundaries, i.e.,

  $$\boldsymbol{u}|_{\Gamma_{\text{fore}}} = \boldsymbol{u}|_{\Gamma_{\text{aft}}}, \ \boldsymbol{\sigma_u n}|_{\Gamma_{\text{fore}}} = \boldsymbol{\sigma_u n}|_{\Gamma_{\text{aft}}},$$

  $$\theta|_{\Gamma_{\text{fore}}} = \theta|_{\Gamma_{\text{aft}}}, \ K\nabla\theta \cdot \boldsymbol{n}|_{\Gamma_{\text{fore}}} = K\nabla\theta \cdot \boldsymbol{n}|_{\Gamma_{\text{aft}}}.$$

The values for the physical constants used in the numerical experiments are taken from Greve and Blatter (2009) and are shown in Table 1.

For all numerical experiments, we extract surface velocities at points from forward solution fields with specified "truth" geothermal heat flux field as synthetic observations, and add random Gaussian noise to lessen the "inverse crime", which





occurs when the same numerical method is used to both synthesize the observations and drive the inverse solution (e.g., Kaipio and Somersalo, 2005). We specify the noise level through the signal-to-noise ratio (SNR), which is defined as the ratio between the average surface velocity $\langle u \rangle$ of $N^{\mathrm{obs}}$ observation points and the standard deviation of the added noise, $\sigma_{\mathrm{noise}}$, i.e.,

$$\mathrm{SNR} = \frac{\langle u \rangle}{\sigma_{\mathrm{noise}}}, \quad \text{with } \langle u \rangle = \sqrt{\frac{1}{N^{\mathrm{obs}}} \sum_{k=1}^{N^{\mathrm{obs}}} \|\boldsymbol{u}_k\|^2}. \tag{41}$$

We choose a regularization parameter that approximately satisfies Morozov's discrepancy principle (Vogel, 2002), i.e., we find a regularization parameter such that $\langle \boldsymbol{u}_\gamma - \boldsymbol{u}^{\mathrm{obs}} \rangle \approx \delta$, where $\delta$ is the noise level and $\boldsymbol{u}_\gamma$ is the surface velocity at the observations points corresponding to the inferred geothermal heat flux for a regularization parameter $\gamma$.

## 5.1 Two-dimensional model problem

First, we consider inversion for a geothermal heat flux in a two-dimensional problem. We discretize the domain, $\Omega$, into
quadrilaterals (Figure 2), and employ biquadratic elements for the velocity components, bilinear elements for pressure, and biquadratic elements for temperature. Linear elements are used for the unknown geothermal heat flux $G$ defined on $\Gamma_b$, unless otherwise specified. For this discretization, the combined number of unknowns for the velocity, pressure, and temperature fields is 2,392, and for the geothermal heat flux it is 41. Unless specified otherwise, we use 50 uniformly distributed observation points on the top surface.

*Inversion for heat flux containing large and small wavelength variations.* We first study inversion with a "truth" geothermal heat flux defined by

$$G(x) = \frac{1}{20} + \frac{1}{20} \exp\left(-\frac{(x - L/2)^2}{2(L/10)^2}\right) + \frac{1}{100} \sin\left(\frac{2\pi x}{L}\right). \tag{42}$$

Here, the second and third term contribute a large and a small wavelength variation to the geothermal heat flux, respectively; the resulting "truth" heat flux is visualized in Figure 4b. The Gauss-Newton algorithm terminates after 8 Gauss-Newton iterations
(requiring a total of 105 CG iterations), when a decrease in the norm of the gradient by a factor of $10^5$ was achieved.

In Figure 3, we show the forward solution (temperature and velocity fields) obtained by solving (6)–(10) with the geothermal heat flux given by (42). This figure shows that the ice is in a cold state, i.e., the temperature is below the pressure melting point. We note that a temperature boundary layer is formed at the base of the ice corresponding to the accumulation zone due to the flow of cold ice from the surface and due to the advection dominating the diffusion. As the warmer ice close to the base flows
toward the surface in the ablation zone, another temperature boundary layer forms at the surface at the right part of the ablation zone.

Figure 4a shows the synthetic pointwise velocity observations, obtained by extracting the surface velocities shown in Figure 3 at the observation points, followed by adding independent noise corresponding to SNR = 20 to each observation. The reconstruction of the geothermal heat flux and the corresponding recovered velocity fields are shown in Figures 4b and 4a,
respectively. Figure 4a illustrates that inversion is able to fit the data to within the noise. However, while the large-wavelength component of the geothermal heat flux is well recovered, the small-wavelength variations cannot be reconstructed. This can





be explained by the fact that the sensitivity of the surface velocity to the small-wavelength variations in $G$ is low, due to the smoothing property of the Stokes operator. These low sensitivities are overwhelmed by the noise in the data, making the reconstruction of the small-wavelength component of $G$ impossible. Taken together, these results reinforce the ill-posedness of the inverse problem.

*Inversion for different SNR and different wavelength variation in the geothermal heat flux.* We continue with a systematic study of the consequence of the wavelength variation of the geothermal heat flux and of the SNR on the reconstruction. For this study, we consider different wavelengths of variations in the "truth" geothermal heat flux

$$G(x) = \frac{3}{50} - \frac{2}{50} \sin\left(\frac{2\pi x}{L_w}\right), \tag{43}$$

where $L_w$ is taken as 80, 40 or 20 km. As before, we solve the forward problem with the true geothermal heat flux field (43) for the different wavelengths. Then we add noise with a given SNR to the resulting point velocity observations and use these synthetic observations to reconstruct the geothermal heat flux field.

In Figure 5, we show inversion results for different wavelength variations and for different noise levels. To assess the reconstruction quantitatively, in Table 2 we report on the relative error between the "truth" and reconstructed geothermal flux fields for various wavelengths variations and noise levels. This relative error is computed as follows:

$$e(G) = \frac{\|G - G_{\text{true}}\|_{L_2}}{\|G_0 - G_{\text{true}}\|_{L_2}}, \tag{44}$$

where $G_0 = 0.06 \text{ Wm}^{-2}$ is the mean of the "truth" geothermal heat flux $G_{\text{true}}$. Based on the results summarized in Figure 5 and Table 2, we make the following observations:

1. For fixed wavelength, the reconstructed geothermal heat flux $G$ approaches $G_{\text{true}}$ as the noise level decreases.

2. For fixed noise level, smaller wavelength variations of the geothermal heat flux are more difficult to reconstruct.

3. For small wavelength (e.g., $L_w = 20$ km, see Figure 5f) and small noise (e.g., SNR = 100), the wave crests and valleys of the truth heat flux are recovered, but the magnitude of the reconstruction is smaller than of the "truth" geothermal heat flux. For the case with larger noise (e.g, SNR = 20), the reconstruction does not detect the crests and valleys, although the corresponding surface velocity still matches the observations within the noise (see the black curve in Figure 5f). The results in Table 2 confirm these findings, in particular we note that the relative errors for the small wavelength and large noise (i.e. $L_w = 20$ km with SNR = 20 or SNR = 10) is roughly 100%, i.e., the reconstruction fails to capture the variations of the "truth" geothermal heat flux.

*Influence of the number of observations and the mesh resolution.* We consider 10, 25, 50, and 100 uniformly distributed observation points and two different meshes, namely a mesh consisting of $40 \times 4$ elements with linear elements for the geothermal heat flux, and a mesh consisting of $80 \times 4$ elements, which uses quadratic finite elements for the unknown heat flux. The "truth" geothermal heat flux field is given by (43) and the noise level in the observations is SNR = 20. In Figure 6a, we show





the "truth" and the reconstructed geothermal heat flux fields for 10, 25, 50 and 100 observation points. This figure shows that the reconstruction improves significantly as the number of observation points increases.

To study the influence of the number of observations and of the mesh resolution on the ill-posedness of the inversion, we study the properties of the Hessian matrix of the data misfit component of $\mathcal{J}$ (i.e., the Hessian of the first term in the cost function defined by (13)). To explain why the Hessian of the data misfit provides insight into the ill-posedness of an inverse problem, consider a Taylor expansion of the data misfit term in $\mathcal{J}$ about the solution of the inverse problem (13). When the inverse solution is able to fit the data to within the noise and the noise is small, the gradient of the data misfit component of $\mathcal{J}$ is negligible, and the local behavior of $\mathcal{J}$ is governed by the Hessian term. Perturbations of the geothermal heat flux in directions associated with large eigenvalues of the data misfit Hessian result in a significant change of the cost functional and are thus well constrained by the data misfit term. On the other hand, the cost functional is not sensitive to perturbations in parameter directions associated with small eigenvalues of the Hessian and, as a consequence, these directions are only weakly (or not at all) constrained. The more such directions exist, the more ill-posed the inverse problem is. Thus, the spectrum of the Hessian provides information on which directions in the parameter space can be reliably recovered (namely, those corresponding to large eigenvalues) and which directions are poorly or not at all recoverable (those corresponding to small or zero eigenvalues). The Hessian also plays an analogous role in quantifying uncertainty in the inversion in the framework of Bayesian inference (Tarantola, 2005). Here, the inverse of the Hessian provides an approximation to the covariance matrix of the posterior probability density function, which is regarded as the solution of an appropriately formulated Bayesian inverse problem. An approximation of the inverse Hessian can be computed even for large-scale inverse problems by exploiting low-rank properties that are typical for many ill-posed inverse problems (Flath et al., 2011; Petra et al., 2014; Kalmikov and Heimbach, 2014).

Since we are using a Newton method to solve the inverse problem, the Hessian matrix is available (or more correctly, its action in a particular direction, as presented in Section 3, is available, and this is all that is required to extract the spectrum using a Lanczos method). In the following, we use the spectrum of the data misfit component of the Hessian to characterize how the ill-posedness of the inverse problem varies with the number of observations and the mesh resolution. In Figure 6b, the spectra of the data misfit Hessians for different numbers of observations and the two different mesh resolutions are shown. If we were to include the regularization in the Hessians (i.e., consider the full Hessian of the cost functional $\mathcal{J}$, not only the data misfit term), these spectra would not collapse to zero but remained bounded from below. Figure 6b shows that the spectra of the data misfit Hessians decay rapidly, in particular for the cases with a small number of observation points. This illustrates the severe ill-posedness of the geothermal heat flux inversion problem considered here. Note that since the observed data are the horizontal and vertical components of the velocity fields at points on the top surface, the rank of the data misfit Hessian cannot be larger than twice the number of the observations points; this can be seen, for instance, in the spectrum for the case with 10 observation points. As the number of observation points increases, the number of nonzero eigenvalues of the data misfit Hessian increases. However, the largest eigenvalues, which correspond to the parameter directions most strongly constrained by the data, do not change as the number of observations increases. Thus, these parameter directions are already well constrained by a small number of observations. Also, note that the finer discretization of the model and the heat flux





becomes more important as more observations are available. This is because the additional information obtained from more observations can be used to better inform the geothermal heat flux only if the finite element discretization of that heat flux and of the model equations has sufficiently many degrees of freedom to capture that information.

## 5.2 Three-dimensional model problem

Next, we consider a three-dimensional model problem with domain $\Omega$ of length and width $L$= 80 km and height given by (39). A cross section of the geometry is shown in Figure 1. We discretize the domain, $\Omega$, using $20 \times 10 \times 2$ hexahedra. The combined number of degrees of freedom for velocity, pressure and temperature is 17,913 and for the geothermal heat flux it is 231. In this numerical experiment, we aim to reconstruct the spatially varying geothermal heat flux

$$G(x,y) = \frac{1}{20} + \frac{1}{20} \exp\left(-\frac{(y-L/2)^2}{2(L/10)^2}\right), \tag{45}$$

where $(x,y) \in [0,L] \times [0,L]$. We use $30 \times 30$ uniformly distributed pointwise velocity observations at the top surface, and add noise to the synthetic observations such that SNR = 20. The algorithm converged after 6 Gauss-Newton iterations (involving a total of 30 CG iterations), where we again terminated the iterations as soon as the norm of the gradient was decreased by a factor of $10^5$.

The top row in Figure 7 shows the velocity observations on the top surface and the surface velocities obtained with the reconstructed geothermal heat flux. The bottom row shows the "truth" and the reconstructed geothermal heat flux fields. We note that the higher geothermal heat flux in the center warms up the ice, results in lower viscosity and, thus, faster ice flow. Also note that the inverse solution is able to fit the reconstructed velocity to the observations to within the noise. The geothermal heat flux in the upstream part is well recovered, but the reconstruction deteriorates downstream. We attribute this phenomenon to the fact that the heating effect mostly affects the downstream surface flow and, hence, the larger heat flux near the outflow boundary has little effect on the ice flow velocity on the surface above. In Figure 8, we show the temperature field based on the reconstructed geothermal heat flux. We note that the ice is cold, with a temperature field comparable to the two-dimensional model problem on each slice (Fig. 8a). In Figure 8b, we show the temperature at the base $\Gamma_b$ of the ice. Note that the temperature is higher in the center due to the non-uniform geothermal heat flux, but it is below the melting point everywhere in $\Gamma_b$.

## 6 Fully-coupled versus one-way coupled approaches in multiphysics inversion

Multiphysics *forward* problems are commonly solved using so-called "one-way coupled" or "operator-split" approaches. For example, for a coupled problem with two physics components, the first physics subproblem would be solved assuming the state variables of the second physics subproblem remain fixed, after which the second physics subproblem is solved using the just-computed first physics state variables. One then iterates until convergence, which is guaranteed only if the spectral radius of a certain iteration matrix is less than unity. If the iteration converges, it converges to the correct solution. Such one-way coupled solvers have been used successfully for ice flow forward problems (Dahl-Jensen, 1989; Hvidberg, 1996; Price et al., 2007; Zwinger et al., 2007; Zhang et al., 2011), in which case the solver iterates back-and-forth between Stokes and energy equation





solves, passing velocities from the former to the latter, and temperatures from the latter to the former. The convergence rate is only linear, as opposed to quadratic for a fully-coupled Newton forward solver, but one might still prefer the one-way coupled approach due to its ability to capitalize on existing single-physics solvers and codes, its avoidance of computing Jacobians of coupling terms, and the difficulties of designing preconditioners for the fully-coupled Jacobian. Therefore, it is tempting to use

the same operator from a one-way coupled forward solver to also solve the adjoint problem during inversion. However, this also leads to an incorrect adjoint operator, since it discards some of the coupling blocks within the operator. This in turn leads to an incorrect gradient, which can lead to inaccurate or incorrect solutions of the inverse problem, depending on how strong the coupling terms in the Jacobian of the fully coupled problem are. In this section we illustrate this issue using the multiphysics inverse problem given by the coupled system consisting of the Stokes equations (1) and (2), and the energy equation (3). In the

rest of this section, to simplify the notation, we drop the $h$ superscripts on discrete variables.

    In the following discussion, we express the forward problem (6)–(10) in terms of the residuals of the discretized equations, as follows:

$$\boldsymbol{r}_u(\boldsymbol{u},\boldsymbol{p},\boldsymbol{\theta}) = 0, \quad \boldsymbol{r}_p(\boldsymbol{u}) = 0, \quad \boldsymbol{r}_\theta(\boldsymbol{u},\boldsymbol{\theta}) = 0,$$

where $\boldsymbol{u}$, $\boldsymbol{p}$, and $\boldsymbol{\theta}$ denote the discretized velocity, pressure, and temperature, respectively, and $\boldsymbol{r}_u$, $\boldsymbol{r}_p$, and $\boldsymbol{r}_\theta$ are the discrete

residuals of the momentum, mass, and energy equations, respectively. The discrete adjoint system corresponding to (19)–(23) can be written as

$$
\begin{bmatrix}
\mathbf{B}_{uu}^T & \mathbf{B}_{up} & \mathbf{B}_{\theta u}^T \\
\mathbf{B}_{up}^T & 0 & 0 \\
\mathbf{B}_{u\theta}^T & 0 & \mathbf{B}_{\theta\theta}^T
\end{bmatrix}
\begin{bmatrix}
\boldsymbol{v} \\
\boldsymbol{q} \\
\boldsymbol{\lambda}
\end{bmatrix}
=
\begin{bmatrix}
\boldsymbol{f}_v \\
0 \\
0
\end{bmatrix},
\tag{46}
$$

where $\boldsymbol{v}$, $\boldsymbol{q}$, and $\boldsymbol{\lambda}$ denote the discretized adjoint velocity, pressure, and temperature, respectively, and $\boldsymbol{f}_v$ is the right hand side of the discrete adjoint momentum equation corresponding to the misfit term in (22). Here, $\mathbf{B}_{uu} = \partial\boldsymbol{r}_u/\partial\boldsymbol{u}$, $\mathbf{B}_{up} = \partial\boldsymbol{r}_u/\partial\boldsymbol{p}$,

$\mathbf{B}_{u\theta} = \partial\boldsymbol{r}_u/\partial\boldsymbol{\theta}$, $\mathbf{B}_{\theta u} = \partial\boldsymbol{r}_\theta/\partial\boldsymbol{u}$, $\mathbf{B}_{\theta\theta} = \partial\boldsymbol{r}_\theta/\partial\boldsymbol{\theta}$. We note that the submatrix $[\mathbf{B}_{uu}^T, \mathbf{B}_{up}; \mathbf{B}_{up}^T, 0]$ is the transpose of the linearized discrete Stokes operator (which is in fact symmetric), and $\mathbf{B}_{u\theta}$ and $\mathbf{B}_{\theta u}$ are Jacobians of the coupling terms between the Stokes and the energy equations. In particular, $\mathbf{B}_{u\theta}$ is the term corresponding to the derivative of the momentum residual $\boldsymbol{r}_u$ with respect to the temperature, and $\mathbf{B}_{\theta u}$ is the term corresponding to the derivative of the energy residual $\boldsymbol{r}_\theta$ with respect to the velocity. We call the gradients obtained when neglecting either of these coupling matrices in the adjoint systems "one-way

coupled gradients" and denote these by $\mathcal{G}_{\text{owc}}$. Next, we study the consequences of the use of one-way coupled gradients on the inversion.

    As an illustration of neglecting Jacobians of coupling terms in the adjoint equation, we neglect $\mathbf{B}_{\theta u}^T$ in (46). Note that we retain $\mathbf{B}_{u\theta}^T$ in the adjoint operator. This allows us to uncouple the adjoint Stokes equation from the adjoint energy equation, leading to a block triangular system. This can be solved by first solving for the adjoint velocity and pressure, and then computing

the adjoint temperature using the just-computed adjoint velocity. Because we have neglected a block within the adjoint operator, we obtain an incorrect adjoint solution, which then leads to an incorrect gradient. How incorrect the gradient is depends on the "magnitude" of $\mathbf{B}_{\theta u}$. To study the implications of using the resulting one-way coupled gradient $\mathcal{G}_{\text{owc}}$ in an inverse problem, we





compare the inverse solution based on the one-way coupled gradient with the solution obtained with the exact gradient $\mathcal{G}_{\text{exact}}$. We summarize our findings in Figure 9. Since the one-way coupled gradient is not the correct gradient of the cost functional $\mathcal{J}$, a descent direction is not guaranteed, and as a result the Gauss-Newton method for solving the inverse problem (13) terminates when the search direction based on the one-way coupled gradient is not a descent direction for $\mathcal{J}$. Thus the solution of the inverse problem based on the one-way coupled gradient can differ from the correct solution not only because the incorrect optimality condition is being satisfied, but also because the search direction can terminate prematurely due to inconsistency of the gradient and cost functional. However, despite the fact that we attempt to solve the wrong optimality conditions (i.e., vanishing of the one-way coupled gradient rather than the exact gradient), and despite the premature termination, one can still obtain a reasonable approximation of the inverse solution. This is depicted in Figure 9a, which shows the inferred geothermal heat flux based both on the exact gradient and on the one-way coupled gradient. As can be seen, they are close to each other.

In Figure 9b we show the convergence of the norm of the gradient for two iterations, one corresponding to the correct gradient, and one corresponding to the one-way coupled gradient. Note that the one-way coupled gradient iteration terminates prematurely after 11 iterations. Figure 9c explores why the one-way coupled iteration terminated early.

First, we plot the angle between the exact gradient $\mathcal{G}_{\text{exact}}$ and the one-way coupled gradient $\mathcal{G}_{\text{owc}}$, i.e,

$$\cos\phi_2 = \frac{\langle \mathcal{G}_{\text{exact}}, \mathcal{G}_{\text{owc}} \rangle_{\Gamma_b}}{\|\mathcal{G}_{\text{exact}}\|_{L^2}\|\mathcal{G}_{\text{owc}}\|_{L^2}}. \tag{47}$$

As can be seen, initially the one-way coupled gradient direction coincides with the exact gradient direction, but the angle between them increases substantially in the later iterations. Beyond this incorrectness, the Gauss-Newton search direction based on the one-way coupled gradient is not even a descent direction for the cost function $\mathcal{J}$, leading to the premature termination of the Gauss-Newton iterations. This is because the one-way coupled gradient is not consistent with the contours of $\mathcal{J}$ (which are computed using the correct forward model). Note that a search direction $\tilde{G}$ is a descent direction only if its angle $\phi_1$ with respect to the negative gradient direction $-\mathcal{G}_{\text{exact}}$ is less than $\pi/2$, i.e., $\cos(\phi_1) > 0$ (Nocedal and Wright, 2006, p. 21). The cosine of $\phi_1$ is thus given by

$$\cos\phi_1 = \frac{\langle -\mathcal{G}_{\text{exact}}, \tilde{G} \rangle_{\Gamma_b}}{\|\mathcal{G}_{\text{exact}}\|_{L^2}\|\tilde{G}\|_{L^2}}, \tag{48}$$

where $\tilde{G}$ is the Newton search direction based on the one-way coupled approach. Figure 9c plots the values of $\cos\phi_1$. As can be seen, the line search fails at iteration 11, when $\cos(\phi_1) < 0$. In other words, not only is the computed search direction incorrect (relative to that of a correct Gauss-Newton step), but it does not even point downhill!

These results illustrate several important characteristics of approximations made in inverse problems governed by multi-physics forward models. First, discarding the Jacobians of coupling terms within the adjoint operator can result in substantially incorrect gradients. This could lead to incorrect solution of the inverse problem, due to the fact that the vanishing of the gradient constitutes the first order necessary condition for solution of the inverse problem. It could also lead to premature termination of the iterations, due to the loss of a descent direction stemming from inconsistency of the gradient with the contours of the cost function. Second, despite the incorrect gradient, it may still be possible to obtain a reasonable solution to the inverse problem,





particularly if the discrepancy between exact and approximate gradients remains small for a sufficient number of iterations to provide a good approximate inverse solution.

## 7    Conclusions

We have formulated an inverse problem for estimating the uncertain geothermal heat flux at the base of an ice sheet or glacier in a thermomechanically coupled nonlinear Stokes model from surface velocity observations. Since the forward problem involves an advection-dominated energy equation, a streamline upwind Petrov-Galerkin (SUPG) stabilization was used to suppresses non-physical oscillations in the temperature field. This required use of a discretize-then-optimize approach to compute adjoint-based gradients and Hessians. We advocated an inexact Newton method to solve the discretized inverse problem. Using two and three-dimensional model problems, we studied the identifiability of the geothermal heat flux field on the basal boundary. We found that the quality of the reconstruction deteriorates with smaller wavelength variations of this heat flux and with increasing noise in the observations. In particular, a geothermal heat flux with a mean value of $0.06 \, \mathrm{Wm}^{-2}$ can be reconstructed accurately from observations that contain 1% noise (SNR = 100) when the wavelength-to-ice-thickness ratio is $\sim$20, and when the observations contain 5% noise (SNR = 20), for a wavelength-to-ice-thickness ratio of $\sim$40. In addition, we studied the influence of the number of observations and the mesh resolution on the reconstruction and found that the reconstruction improves significantly as the number of observation points increases, provided the discretization is fine enough.

Moreover, we derived expressions for the gradient and the Hessian of the cost functional for a fully thermomechanically coupled Stokes forward model. We discussed problems that can occur when the gradient is approximated by a so-called one-way coupled approach, in which the two-way coupling of Stokes and the energy equations is replaced by one-way coupling, as is frequently done within forward solvers. The results show that the inversion based on a one-way coupled approach can fail to converge due to the inconsistency of the gradient and the cost functional, leading to the loss of a descent direction. Nevertheless, one might still obtain a reasonable approximate inverse solution, particularly if important features of the reconstructed solution emerge early in optimization iterations, before the iterations terminate prematurely.

We have used synthetic observations on idealized geometries to probe the limits of invertibility for the geothermal heat flux field. We have assumed that the ice is cold everywhere and thus enforced a no-slip boundary condition at the base. In reality, the ice may reach the pressure melting point at some basal locations. This requires a different set of boundary conditions, which account for ice either below or at the melting point. Solution of thermomechanically coupled ice flow models with such variational inequality boundary conditions is the subject of our current work.

## Appendix A:  Newton's method for the forward problem

Here we briefly discuss the solution of the forward problem (6)–(10) using Newton's method. We again omit $h$ superscripts on discrete variables. To determine an initial guess $\boldsymbol{u}_0, p_0, \theta_0$ for the Newton solver, we first solve for the velocity and pressure $\boldsymbol{u}_0, p_0$ by setting $\theta(x,y) = \theta_s(x)$, and compute the stabilization factor $\tau = \tau(\boldsymbol{u}_0)$ for the SUPG method. Then, we solve for the



initial temperature $\theta_0$ using $\boldsymbol{u}_0$. Finally, we take $(\boldsymbol{u}_k, p_k, \theta_k)$ as initial guess and solve $(\boldsymbol{u}_{k+1}, p_{k+1}, \theta_{k+1})$ from the coupled system simultaneously. All the above nonlinear solves are done using Newton's method and choosing a step length such that the residual of the forward problem is sufficiently decreased at each iteration.

*Acknowledgements.* We appreciate helpful comments from Ginny Catania. This work was partially supported by NSF's Cyber-Enabled Discovery and Innovation Program (OPP-0941678) and DOE Office of Science, Office of Advanced Scientific Computing Research (DE-SC0002710, DE-SC0009286). H. Zhu also acknowledges funding through the ICES NIMS Fellowship.





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





**Table 1.** Notation, units, and values for parameters and constants. Shown are Glen's flow law exponent $n$, the pre-exponential constant $A_0$, the activation energy $Q$, the universal gas constant $R$, the gravitational acceleration $g$, the heat capacity of ice $c$, the thermal conductivity $K$, and the density $\rho$ of ice. Note that we choose $Q$ and $A_0$ for the case that the temperature of the ice is below -10°C, as the solutions are mostly within this range.

| Parameter | Value | SI Unit |
|-----------|-------|---------|
| $n$ | 3 | - |
| $A_0$ | $3.985 \times 10^{-13}$ | $\mathrm{Pa}^{-3}\mathrm{s}^{-1}$ |
| $Q$ | $6 \times 10^4$ | $\mathrm{J(mol)}^{-1}$ |
| $R$ | 8.314 | $\mathrm{J(molK)}^{-1}$ |
| $g$ | 9.81 | $\mathrm{ms}^{-2}$ |
| $c$ | 2009 | $\mathrm{J(kgK)}^{-1}$ |
| $K$ | 2.10 | $\mathrm{W(mK)}^{-1}$ |
| $\rho$ | 910 | $\mathrm{kgm}^{-3}$ |

**Table 2.** The relative error $e(G)$, computed using (44), between the "truth" (43) and the reconstructed geothermal flux for wavelength variations $L_w = 80$, 40 and 20 km and for signal-to-noise ratios SNR = 100, 20 and 10.

| $L_w$ | SNR | | |
|-------|-----|-----|-----|
| | 100 | 20 | 10 |
| 80 | 0.038 | 0.136 | 0.266 |
| 40 | 0.136 | 0.570 | 0.938 |
| 20 | 0.528 | 0.999 | 1.002 |





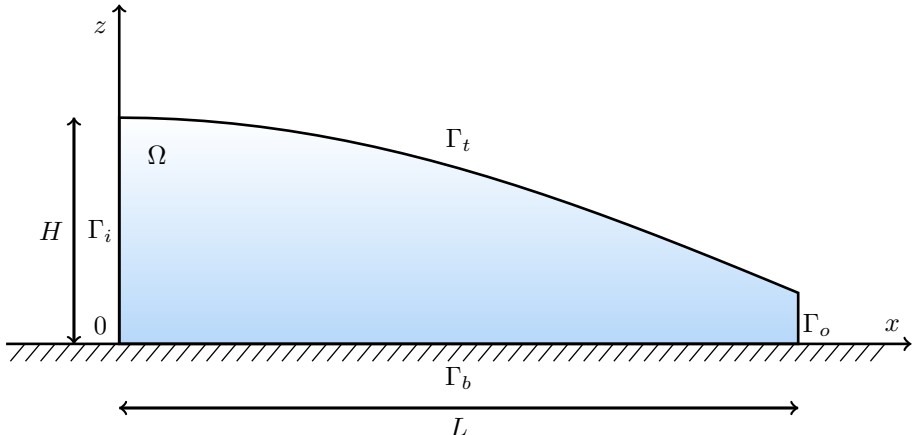

**Figure 1.** Coordinate system and cross section through a three-dimensional slab of ice, as used in the computational experiments (exaggerated in height for visualization).

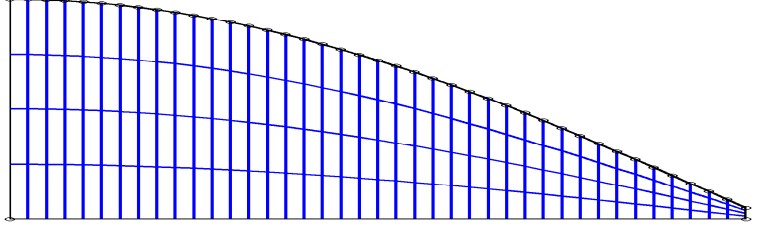

**Figure 2.** Two-dimensional mesh (exaggerated in height for visualization).

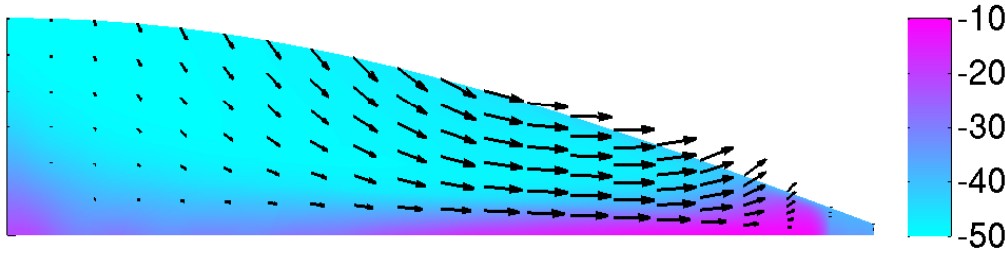

**Figure 3.** Temperature and velocity found by solving the forward problem with geothermal heat flux given in (42). The color visualizes the temperature (in °C) and the arrows show the corresponding velocity field.





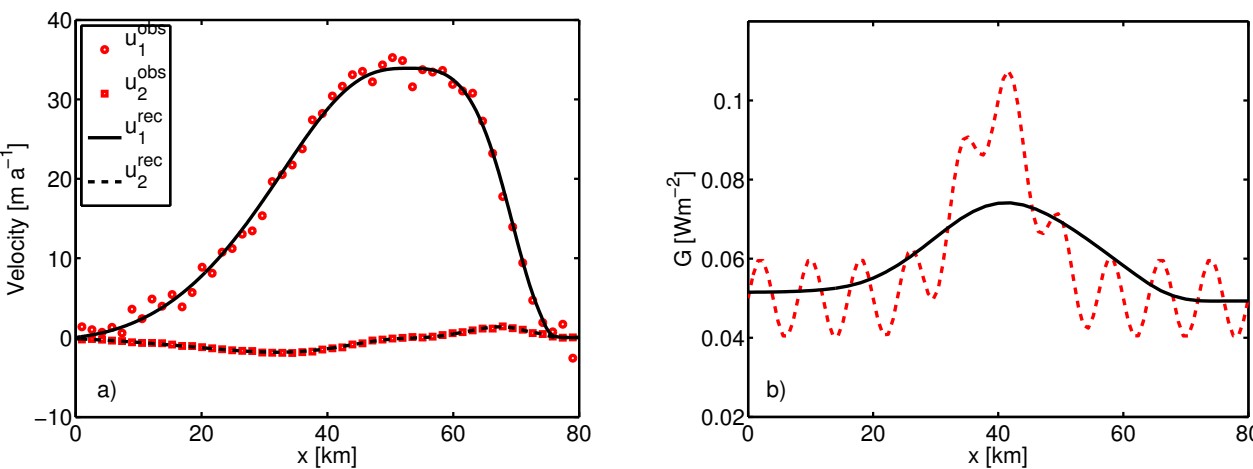

**Figure 4.** Reconstruction of geothermal heat flux $G$ in two-dimensional model problem with SNR = 20. (a) surface velocity observations (red dots show horizontal component; red squares vertical component) and reconstructed velocities (black solid line shows horizontal component; black dashed line shows vertical component); (b) "truth" and reconstructed geothermal heat flux (the dashed line shows the "truth" geothermal heat flux defined in (42); the solid line shows the reconstructed geothermal heat flux).





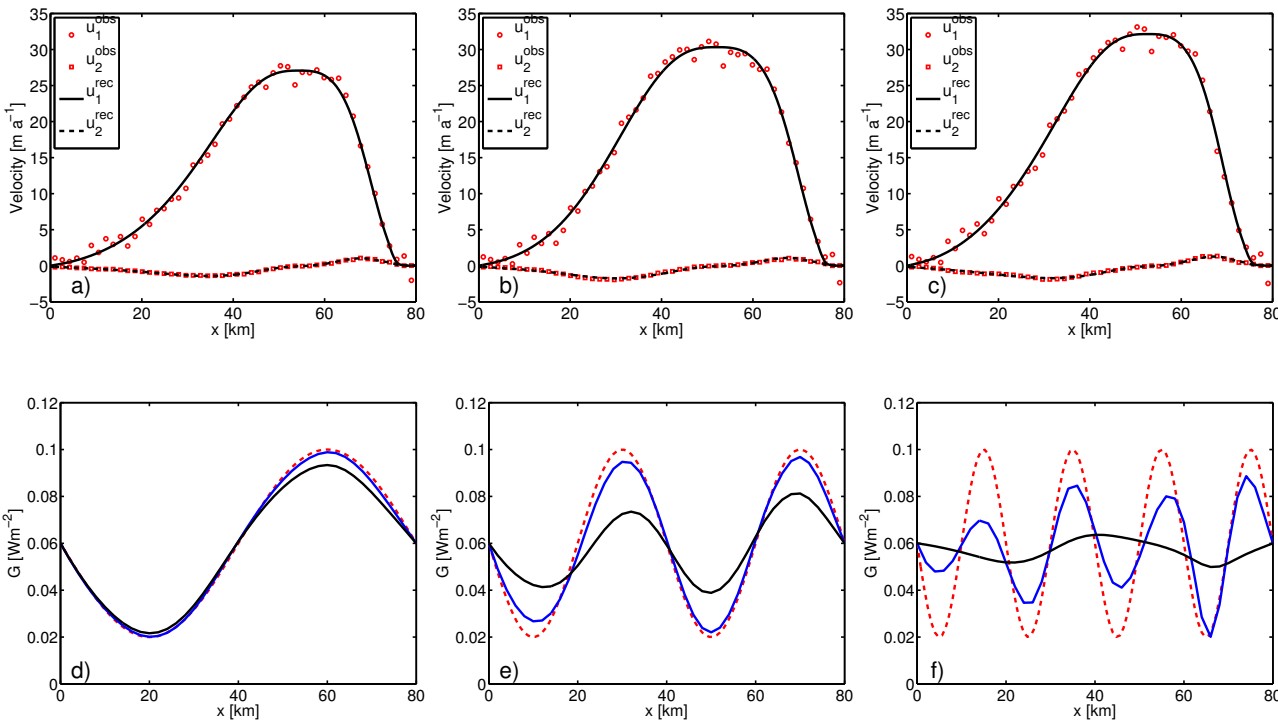

**Figure 5.** Reconstructions of geothermal heat flux with different noise levels and different wavelength variations for the two-dimensional model problem. In (a), (b) and (c) we display synthetic observations (red circles for horizontal component and red squares for vertical component; the data correspond to SNR = 20) computed from the "truth" heat fluxes $G_{\text{true}}$ with $L_w = 80$, $L_w = 40$, and $L_w = 20$ shown as red dashed lines in (d), (e) and (f), respectively. Also shown in (a), (b) and (c) are the velocity components (solid and dashed black lines) corresponding to the reconstructed heat fluxes shown as black solid lines in (d), (e) and (f). In (d), (e) and (f) we additionally show the reconstruction of the geothermal heat fluxes from surface velocity data with SNR = 100 (blue solid lines).



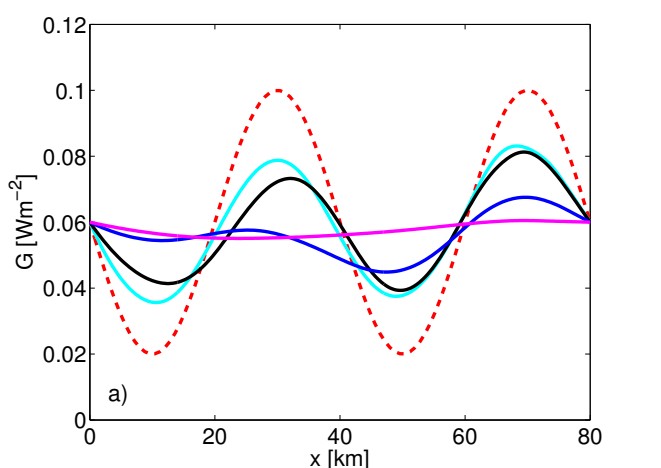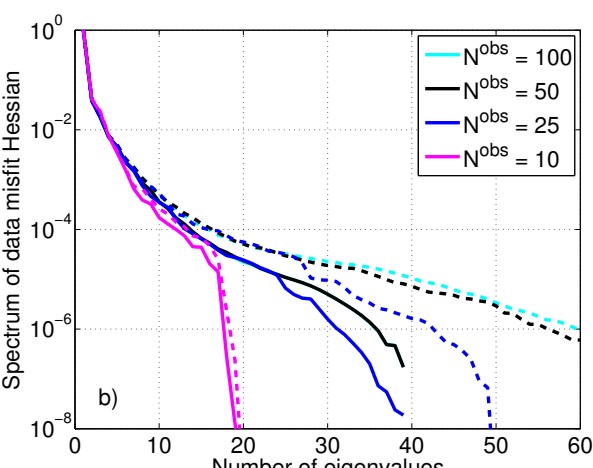

**Figure 6.** Shown in (a) are the "truth" (red dashed line) and the reconstructed geothermal heat flux fields for 10 (magenta), 25 (blue), 50 (black), and 100 (cyan) uniformly distributed observation points. These reconstructions are computed using the finer discretization, which uses 80 quadratic elements for the heat flux; the reconstructions obtained with the coarser discretization are similar. In (b), the corresponding (normalized) spectrum of the data misfit Hessian for the two different discretizations (solid lines correspond to the $40 \times 4$ element mesh with a linear basis for the heat flux, and dashed lines correspond to the $80 \times 4$ element mesh with a quadratic basis for the heat flux) and different numbers of observation points are shown. We note that the cyan solid line is covered by the black solid line as the recoverable information is limited by the mesh resolution.





**Figure 7.** Reconstruction of geothermal heat flux $G$ for the three-dimensional model problem. Shown in (a) are observations of the surface velocity (arrows and contour lines) with SNR = 20, and (b) shows the surface velocity corresponding to the reconstructed geothermal heat flux. In (c), we show the "truth" geothermal heat flux defined in (45), and (d) shows the reconstructed heat flux.




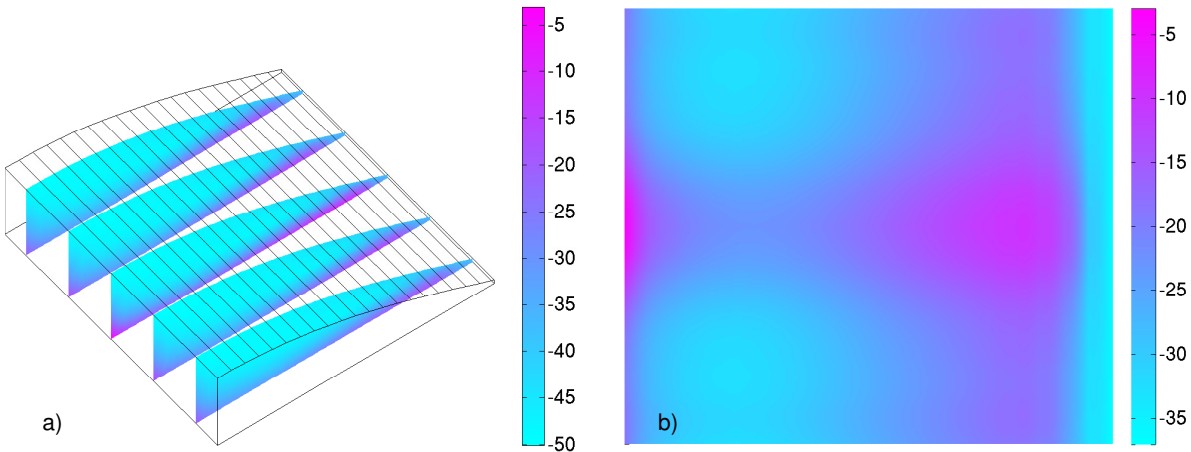

**Figure 8.** The temperature field (in °C) corresponding to the reconstructed geothermal heat flux $G$ for the three-dimensional model problem. Shown in (a) are slices through the domain, and (b) shows the temperature at the base $\Gamma_b$.

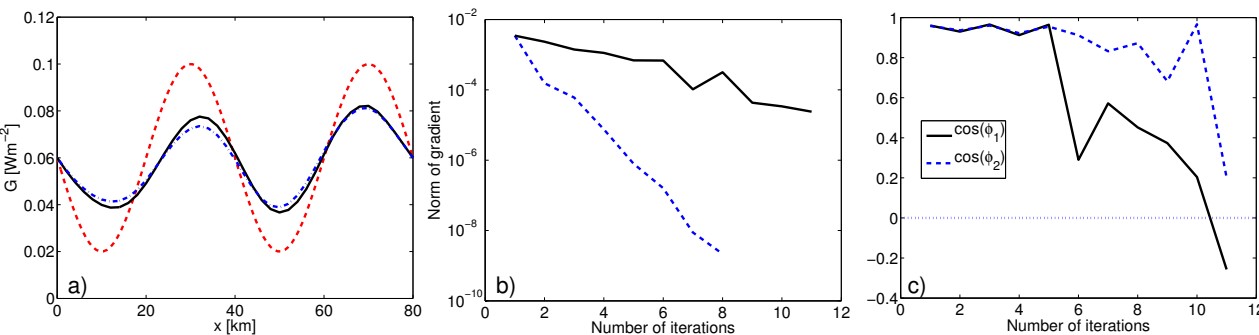

**Figure 9.** Reconstructions of the geothermal heat flux based on the one-way coupled gradient obtained when the coupling matrix $\mathbf{B}_{\theta u}^T$ is neglected in the adjoint system for observations with SNR = 20. Shown in (a) is the "truth" geothermal heat flux $G_{\text{true}}$ (red dashed line), the reconstructions $G$ based on the exact gradient (blue dash-dot line), and based on the one-way coupled gradient (black solid line). In (b), the norm of the gradients during the iteration is shown. The blue dashed line shows the convergence of the inexact Gauss-Newton algorithm when the exact gradient is used. The black solid line show the convergence for the Gauss-Newton method based on the one-way coupled gradient, i.e., the norm of the exact gradient in each iteration. For the iteration using the one-way coupled gradient, we show in (c) the cosine of the angle between the search direction and the steepest descent direction (black solid line; see (48)) and the cosine of the angle between the one-way coupled gradient and the exact gradient in each iteration (blue dashed line; see (47)). When the former value becomes negative, the search direction is not a descent direction and the algorithm terminates as the line search fails.