# Peer review of "Inversion of geothermal heat flux in a thermomechanically coupled nonlinear Stokes ice sheet model"

_The Cryosphere, 2015_

## Referee Comment (RC1) · Anonymous Referee #1 · 11 Feb 2016

**Review of Zhu et al. Manuscript**

This manuscript presents a new model that carry out inversions for the basal geothermal heat flux from surface velocity observations. This is the first piece of work I am aware of to present such a model, and does so very clearly. The manuscript is extremely well written and I would say it is almost ready to be published as it is. Not being an expert in numerics I am probably not the best person to assess the discussion in section 4, but hopefully another reviewer can analyse this section more carefully.

The one major thing I question about this work is the applicability of it to real data. Given uncertainties surrounding other parameters, could such an inversion really give us sensible predictions for the geothermal heat flux? This question does not take away from the interest of the paper as a mathematical exercise, but since it is for a cryospheric journal it would be good if the potential of the model (or lack of) was discussed a bit further in this paper.

To address this point I suggest you add in a section before the conclusions about applicability of the method and future work. You say in the first paragraph of the introduction that you want to study the prospects for, and limitations of, inferring the geothermal heat flux form surface ice velocities, but where really is any discussion of this? Your assessment of the ability to invert depending on length scales of heat flux and the noise level in velocity observations is obviously very relevant but how does this correspond to what we expect from real data?

**Minor comments**

- **Abstract, lines 17-19** Long sentence. Split into two.

- **Page 6, line 15**. Aren't there some more recent estimates for geothermal heat flux in certain areas of Antarctica? e.g. Fisher et al, 2015, Science.

- **Equations 19-23** Could we have a table of variables etc to reference? Had to spend some time going back to remind myself of what e.g. $\mathcal{B}^*$ is.

- **Equations 19-23** explain why some terms in blue.

- **page 15, line 5** Figure 3 referenced before Figure 2 (page 16, line 10). Swap order of figures.

- **Figure 4** Legend label overlapping with surrounding box.

---

## Referee Comment (RC2) · D. Brinkerhoff (Referee) · 16 Feb 2016

This paper deals with the difficult inverse problem of inferring the geothermal heat flux under an ice sheet under strict assumptions of non-slip and cold basal conditions. The authors derive the infinite-dimensional forms required for generation of a Newton's method. The gradient and Hessian of the proposed least-squares objective function are fully (thermomechanically) coupled, which is an advance over work which has been done previously, which often uses so-called incomplete adjoints. The paper explores the limiting resolution at which geothermal heat flux can be recovered given assumptions of data density and uncertainty. Additionally, the authors explore the effects of using an operator-splitting approach for the adjoint problem.
[Figure]

Major Comments:

While this paper effectively makes its point regarding the numerics of the problem in question, I don't think that it provides enough glaciological relevance to be published in the Cryosphere as is. Thus I propose two options: first, that the paper be resubmitted to Geoscientific Model Development, where the paper may get the appreciation it deserves based more solely on its mathematical merit, or second, that a substantive discussion of the possible implications that this paper's results might have for practical glaciology be added. Some examples of the latter might be the addition of a section that discusses whether, given estimates of error in contemporary remotely-sensed datasets, this method has any promise towards use on real ice masses. Obviously, the methods presented in the paper are limited to cold conditions. Where might these assumptions be valid? What additional factors might complicate the analysis in the real world? Furthermore, the work presented here is done at a rather low resolution. Is this a result of computational efficiency, and would this be a major limiting factor with respect to inverting for heat flux in real glaciers?

Minor Comments:

Generally: I think the paper could benefit from some compaction, both at a small scale (e.g. eliminating as many unnecessary subjective adverbs, such as "significantly", as possible) and also at a large scale (some sections read a little bit like a textbook, cf. P19 L6).

P1 L13: "small wavelength is ambiguous, consider "short-wavelength" instead.

P2 L2: Stokes' equations are always "full," else not Stokes' equations.

P2 L2: It's clear that the model is coupled, so the word "multiphysics" isn't relevant (here and elsewhere).

P2 L2-7: Consider transposing this first paragraph with the second. As it stands, the (brief) literature review splits the problem description.

P2 L8-18: The authors might consider addressing the importance of geothermal heat in determining whether the bed is frozen or not at a given location, which has implications for ice dynamics (via sliding) that very likely exceed rheological effects.

P2 L31: ill-posed is redundant in this line.

P2 L31: The continuous problem is also a PDE-constrained optimization problem, it's just being discretized. Maybe reword to make this more clear.

P2 L32: "Only hope" is a bit strong. See for example Pollard and DeConto (2011) "A simple inverse method for the distribution of basal sliding coefficients under ice sheets, applied to Antarctica" for a counter-example.

P3 L5: The second sentence in the paragraph restates the first, and is not more precise.

P3 L13-35: These sections are a bit too hand-wavey. If the conclusions about one-way versus two-way coupling are solely a result of the work contained herein, this discussion should not be attempted with such depth prior to presenting evidence for said results. If this information is generally understood and valuable as context for the work to be described herein, references should be added.

P4 L12: The statement that "Section 7 provides concluding remarks" is not necessary.

P4 L14-18: All of this information has already been clearly stated in the introduction and may not be necessary here.

P4 L19: Perhaps just "ice can be modelled as" is sufficient here.

P4 L20: "Conservation of" rather than "balance of" seem more appropriate.

P5 L2: Is simultaneous inversion of the flow law exponent and geothermal heat flux undesirable? If so, state as much.

P5 L28: Since this is primarily a glaciology journal, rather than a mathematical one,

please provide a basic reference when talking about weak forms.

P7 L9: "i.e." not necessary.

P8 L12: Clarify that the test functions are acting as Lagrange multipliers.

P8 Eq. 22: I would like to see a little bit more information about $\mathcal{B}$. It is reasonably obvious how it can be constructed (i.e. after discretization evaluating the basis functions at the locations of the observation points). However, since the least-squares misfit isn't integrated over $\Gamma_t$, I fail to see how by equating it with something that is (namely the adjoint stress) can allow the strong form to be recovered. It seems to me that the author's are performing an additional integration of $\Gamma_t$ that is not being shown in the paper, and I would like to see some additional specificity here.

P8 L30: Consider using underbraces to specify what part of the adjoint stress comes from where (and what gets omitted when using the linearized approximation). This is interesting because it becomes clear that there exists an additional term derived from the thermomechanically coupling term that I have never seen considered before.

P9 L3: Throughout the work, the authors use several different notational conventions to refer to effective strain-related quantities (i.e. $\dot{\epsilon}_{II}$, $\dot{\epsilon} : \dot{\epsilon}$, $\mathrm{tr}(\dot{\epsilon}^2)$). It would be better to use a consistent notation throughour (I would favor either the invariant or colon notation, but not the trace notation).

P9 L22: Without stating what this new functional is, I can't really determine if the computed variations are correct. Even referring back to Petra (2012), I can't tell what this Lagrangian on the gradient referred to in the text actually looks like. Is it *only* the forward and adjoint equations imposed via new Lagrange multipliers, or is there a term to be minimized as well? It's not clear from the text.

P10 L5: The incremental forward problem seems to also resemble the adjoint problem, particularly in the changes to the viscosity term relative to the self-adjoint case. Is there a more compelling reason why the forward and incremental forward problems should

be more closely related than the forward and adjoint problems?

P10 L25: Same question as above.

P11 L7: Please make clearer from the outset that the content of this paragraph is paraphrased from Petra (2012), and that a more thorough discussion of the topic is included there.

P11 L10: Can you cite a reference showing why neglecting terms involving the adjoint variable guarantees positive-definiteness in the Hessian? Petra (2012) did this as well, but similarly did not cite a reason for why this should be true.

P11 L29: Check out the "citet" command in bibtex to get citations that look like "Isaac et al. (2015)", rather than "(Isaac et al., 2015)".

P12 L4–6: This front matter is redundant.

P12–13 Sec. 4.1: The specification of the finite element spaces splits the discussion about streamline upwinding. Consider transposing the first and second paragraphs.

P13 L20: The discussion of Galerkin vs. non-Galerkin spaces requires a reference.

P14 L7: This point was already stated in the previous paragraph.

P14 L13: Since the adjoint temperature isn't used here except as a means to facilitate optimization, is the lack of equivalency in the small mesh limit between the discrete and continuous (rather, non SUPG) temperature all that meaningful?

P14 L17: This sentence provides no information.

P14 L21–27: I'm not sure that this is true; the authors spent the previous paragraph talking about how the advantage of OTD is that one can use stabilization terms that vanish as the residual does, but that this creates other problems. This section would seem to suggest (wrongly) that it is DTO that has this property. I could of course be wrong, in which case the authors might humor me by re-writing this section more

transparently.

P14 L32: Perhaps "We consider both a two-dimensional flowline case and a three-dimensional map plane case" would be more clear.

P14 L33: $s(x)$ is usually reserved for surface elevations in glaciological parlance, where $H(x)$ is reserved for ice thickness. In these examples the two appear to coincide because of the flat bed, but this could become confusing for other examples.

P14 L33: Why this particular choice of geometry? There are analytical solutions that are somewhat more realistic than a cosine, such as the Vialov profile. I don't think it matters much with respect to the results, but I think that the choice should be elaborated upon a little bit.

P16 L5: This could be explored with a little bit more depth. Throughout the remainder of the work it seems as though the methods presented herein are selecting overly smooth solutions. Could this be a result of using too-aggressive regularization? Additionally, $\gamma$ computed via Morozov's discrepancy principle would be different for each problem. Was it computed independently for each example, and how did it vary? Would using a smaller value of $\gamma$ allow for better resolution of the small scale variability in geothermal heat flux or is the lack of detail indeed a result of the smoothing nature of Stokes' flow?

Figure 3: The chosen colormap is a little difficult to read. Try something higher contrast.

P17 L2: Once again, I'm not sure whether these results are showing the limits of data recoverability or if they're showing the choice of regularization parameter. A bit more exploration of the latter topic would help to clear this up.

Figure 6: I cannot tell the difference between cyan and blue in this figure.

P19 L6: I am extremely skeptical of the capacity for a model with 2 elements (quadratic or not) to accurately capture the vertical structure of the temperature field, particularly when said temperatures are subject to boundary layers (as the authors noted). Is it possible that this low vertical resolution is contributing to some of the error in the
recovered solution? I suppose that since the same model is being used to generate the surface velocities that this effect would be lessened, but then that brings up the additional issue of whether the overly simplified temperature fields induced by the low resolution are leading to an easier job of recovering the geothermal heat field.

Figure 9: Add legends to (a) and (b). Also, it would be useful to see an additional plot of the cosine of the angle between the two-way coupled descent direction and the exact gradient. Perhaps we would see some "uphill traffic" there as well.

Appendix: Probably not necessary to include this; a reference to any numerical analysis text ought to be fine.

---

## Author Response (AR1)

**Reply to Referee 1:**

Many thanks for your careful reading and your helpful comments and suggestions. Please find below point-by-point replies (in black) to your comments and questions (which are reprinted in blue). To give you an overview of all the changes in the paper, we also provide a diff-document that highlights the changes between the initial submission and this re-submission.

**Major Comments:**

 This manuscript presents a new model that carry out inversions for the basal geothermal heat flux from surface velocity observations. This is the first piece of work I am aware of to present such a model, and does so very clearly. The manuscript is extremely well written and I would say it is almost ready to be published as it is. Not being an expert in numerics I am probably not the best person to assess the discussion in section 4, but hopefully another reviewer can analyse this section more carefully.

We thank the reviewer for appreciating our work and the effort that went into it.

The one major thing I question about this work is the applicability of it to real data. Given uncertainties surrounding other parameters, could such an inversion really give us sensible predictions for the geothermal heat flux? This question does not take away from the interest of the paper as a mathematical exercise, but since it is for a cryospheric journal it would be good if the potential of the model (or lack of) was discussed a bit further in this paper. To address this point I suggest you add in a section before the conclusions about applicability of the method and future work. You say in the first paragraph of the introduction that you want to study the prospects for, and limitations of, inferring the geothermal heat flux form surface ice velocities, but where really is any discussion of this? Your assessment of the ability to invert depending on length scales of heat flux and the noise level in velocity observations is obviously very relevant but how does this correspond to what we expect from real data?

The main point of our paper is the presentation of a (to the best of our knowledge) new formulation for inference of the basal geothermal heat flux from velocity data, and the presentation of a scalable algorithm for the solution of this problem. We have extended a paragraph on the applicability of the method and the assumptions we are making on page 2. While we use model problems in this present paper, the computational methods are designed such that they will scale to realistic large-scale problems and work, in principle, with real data. Note that while being simple, the model problems we use are realistic in terms of dimensions and geometry, constitutive parameters and boundary conditions. While real world problems are without a doubt more complex and challenging, we believe that our model problems are useful and the qualitative results regarding the inversion of geothermal heat flux variations are relevant also for problems with real data and geometry.

**Minor Comments:**

2. Abstract, lines 17-19 Long sentence. Split into two.

In the revised manuscript, we have split this sentence into two.

**3.** Page 6, line 15. Aren't there some more recent estimates for geothermal heat flux in certain areas of Antarctica? e.g. Fisher et al, 2015, Science.

We thank the reviewer for pointing out this reference. We incorporated it into our introduction.

4. Equations 19-23 Could we have a table of variables etc to reference? Had to spend some time going back to remind myself of what e.g.  $B^*$  is.

In the revised manuscript, we have added a new table (see Table 1) that lists the variables used in the forward and adjoint problems. We have also added a description of the adjoint operator  $B^*$  below Equation (25).

5. Equations 19-23 explain why some terms in blue.

Does the reviewer refer to Equation 31-34? The blue terms in these equations correspond to terms in the Hessian expression that involve the adjoint variable, which are neglected for a Gauss-Newton approximation of the Hessian. We explain this on page 11, lines 6-14.

6. page 15, line 5 Figure 3 referenced before Figure 2 (page 16, line 10). Swap order of figures.

We removed the first (unnecessary) reference to Figure 3 such that now they appear in the correct order.

7. Figure 4 Legend label overlapping with surrounding box.

The legend box has been removed from both Figure 4 and Figure 5.

**Reply to Referee 2:**

Many thanks for your careful reading and your helpful comments and suggestions. Please find below point-by-point replies (in black) to your comments and questions (which are reprinted in blue). To give you an overview of all the changes in the paper, we also provide a diff-document that highlights the changes between the initial submission and this re-submission.

**Major Comments:**

1. This paper deals with the difficult inverse problem of inferring the geothermal heat flux under an ice sheet under strict assumptions of non-slip and cold basal conditions. The authors derive the infinite-dimensional forms required for generation of a Newton's method. The gradient and Hessian of the proposed least-squares objective function are fully (thermomechanically) coupled, which is an advance over work which has been done previously, which often uses so-called incomplete adjoints. The paper explores the limiting resolution at which geothermal heat flux can be recovered given assumptions of data density and uncertainty. Additionally, the authors explore the effects of using an operator-splitting approach for the adjoint problem.

While this paper effectively makes its point regarding the numerics of the problem in question, I don't think that it provides enough glaciological relevance to be published in the Cryosphere as is. Thus I propose two options: first, that the paper be resubmitted to Geoscientific Model Development, where the paper may get the appreciation it deserves based more solely on its mathematical merit, or second, that a substantive discussion of the possible implications that this paper's results might have for practical glaciology be added. Some examples of the latter might be the addition of a section that discusses whether, given estimates of error in contemporary remotely-sensed datasets, this method has any promise towards use on real ice masses. Obviously, the methods presented in the paper are limited to cold conditions. Where might these assumptions be valid? What additional factors might complicate the analysis in the real world? Furthermore, the work presented here is done at a rather low resolution. Is this a result of computational efficiency, and would this be a major limiting factor with respect to inverting for heat flux in real glaciers?

We have added some discussion on the "frozen boundary condition" on page 2 and explicitly discuss our assumptions and the reason behind these assumptions. To the best of our knowledge, this is the first paper to propose inferring the geothermal heat flux from flow velocity of the ice sheet. Thus, we believe it is entirely justified to present model problem examples in order to study sensitivities and prospects and limitations of this inverse problem. The surface velocity data that would be used for these inversions is rather accurate (Rignot et al., *Science*, 2011), so it is unlikely that this would be a limiting factor for the method. While we use model problems in this present paper, the computational methods are designed such that they will scale to realistic large-scale problems. The use of model problems is **not** a limitations and methods should first be understood and analyzed using model problems (e.g., Petra, Zhu, Stadler, Hughes & Ghattas, *Glaciology*, 2012), before developing large-scale high-resolution implementations that incorporate realistic geometries and real inversion data (e.g., Isaac, Petra, Stadler and Ghattas, *JCP* 2015).

2. Generally: I think the paper could benefit from some compaction, both at a small scale (e.g. eliminating as many unnecessary subjective adverbs, such as "significantly", as possible) and also at a large scale (some sections read a little bit like a textbook, cf. P19 L6)

We agree that some authors overuse subjective adverbs such as "significantly" and have eliminated two occurrences and replaced one by "substantially". With respect to the remaining three occurrences we believe that it is important to point out that the effect is in fact "significant" and thus the use of the adverb is necessary and justified. We also made sure that no other unnecessary subjective adverbs (e.g., "very") are used anywhere in our paper.

We do not find P19, L6 textbook-like, since this is simply part of the problem description. Did you intend to point to a different part of the paper?

3. P1 L13: "small wavelength" is ambiguous, consider "short-wavelength" instead.

We changed "small/large wavelength variation(s)" to "short-/long-wavelength variation(s)".

4. P2 L2: Stokes' equations are always "full," else not Stokes' equations.

We use the expression "Full Stokes" from the ISMIP-HOM benchmark (Pattyn et al. 2008) and the book *Dynamics of Ice sheets and Glaciers* (Greve and Blatter 2009) to emphasis no simplication is made to the Stokes equations for the solution. We do agree that this terminology is ice-specific and uncommon in classical continuum mechanics, but we use it here as we believe that many readers of "The Cryosphere" have a glaciology background.

5. P2 L2: It's clear that the model is coupled, so the word "multiphysics" isn't relevant (here and elsewhere).

We have removed the term "multiphysics" on P2 and a few more times in the paper when it wasn't crucial. However we kept it in the discussion of issues that can arise due to the use of one-way coupled gradients. For this discussion, the geothermal inverse problem is just one example of a broader class of inverse problems that are governed by a multiphysics forward problem, and our intention is highlight the possible perils of ignoring two-way coupling in the Jacobian. Thus the use of the term "multiphysics" indicates wider applicability.

6. P2 L2-7: Consider transposing this first paragraph with the second. As it stands, the (brief) literature review splits the problem description.

We prefer to keep the original flow of the Introduction section, because in the first paragraph we want to provide motivation at a higher level. In that paragraph we define the problem of interest, challenges, and goals, i.e., establish general context and importance. In the second paragraph, we narrow down the problem, establish specific context, and discuss the relevant research literature.

 P2 L8-18: The authors might consider addressing the importance of geothermal heat in determining whether the bed is frozen or not at a given location, which has implications for ice dynamics (via sliding) that very likely exceed rheological effects.

We now remind the reader of this dependence in that paragraph.

8. P2 L31: ill-posed is redundant in this line.

Since this is the first time in the main text of the manuscript where we say that the inverse problem is ill-posed, we think that it is important to keep the term, especially for readers not familiar with the characteristics of an inverse problem.

9. P2 L31: The continuous problem is also a PDE-constrained optimization problem, it's just being discretized. Maybe reword to make this more clear.

We reworded that sentence to make it easier to read.

**10.** P2 L32: "Only hope" is a bit strong. See for example Pollard and DeConto (2011) "A simple inverse method for the distribution of basal sliding coefficients under ice sheets, applied to Antarctica" for a counter-example.

Our use of "only hope" here is in connection with *efficient* solution. Of course if one has infinite computing time available, one can use a non-derivative method such as simulated annealing to solve an inverse problem; however a gradient-based method will be orders of magnitude faster. Nevertheless, we have changed the phrase from "only hope" to "best hope." We acknowledge that for certain inverse problems (possibly for the problem of estimating the basal sliding co-efficient studied in Pollard and DeConto), ad-hoc/tuning methods can be used. However, the parameter field updates underlying these methods often end up effectively using approximate derivative/sensitivity information; in such cases our statement is also correct. We mainly want to emphasize that some kind of derivative information is crucial for efficient solution.

11. P3 L5: The second sentence in the paragraph restates the first, and is not more precise.

In the second sentence we add concrete details about the kind of inversion tests we perform. Hence, we prefer to keep both sentences.

12. P3 L13-35: These sections are a bit too hand-wavey. If the conclusions about one-way versus two-way coupling are solely a result of the work contained herein, this discussion should not be attempted with such depth prior to presenting evidence for said results. If this information is generally understood and valuable as context for the work to be described herein, references should be added.

We believe that difficulties arising in the one-way coupled approach for multiphysics inverse problems are not generally understood. In lines 21-35 we provide an intuitive discussion of the issue, since we think is suitable for an introduction. Later in the paper, we make our statements precise using the inverse problems studied in this paper to exemplify an inverse problem governed by a multiphysics forward model. Our intention is to highlight that when a one-way coupled approach to computing gradients is used, the performance of the numerical method can deteriorate or, even worse, is not guaranteed to produce the correct solution.

13. P4 L12: The statement that "Section 7 provides concluding remarks" is not necessary.

We have removed that sentence.

14. P4 L14-18: All of this information has already been clearly stated in the introduction and may not be necessary here.

We have removed most of the introduction to this section.

15. P4 L19: Perhaps just "ice can be modelled as" is sufficient here.

We replaced "Ice sheets and glaciers can be modeled" with "Ice can be modeled".

16. P4 L20: "Conservation of" rather than "balance of" seem more appropriate.

"Balance of" is the correct continuum mechanics term, since in general the time rate of change of a quantity (momentum, energy, mass) is balanced by internal and boundary sources. There is no mass source, so the use of "conservation" is correct for that equation. However the linear momentum and energy equations do include source terms, and thus "conservation" is not appropriate in those contexts. Our use of "balance of" is consistent with that of *Theoretical Glaciology*, *Mathematical Approaches to Geophysics* by Hutter (1983).  P5 L2: Is simultaneous inversion of the flow law exponent and geothermal heat flux undesirable? If so, state as much.

It is not undesirable, since the ice rheology law is also uncertain. We replaced the "Instead" with "However" to make it clear that we do not want to suggest that this is undesirable.

18. P5 L28: Since this is primarily a glaciology journal, rather than a mathematical one, please provide a basic reference when talking about weak forms.

We added a reference to Hughes (2000) when first talking about the weak form and the finite element method on page 5.

19. P7 L9: "i.e." not necessary.

This has been removed in the revised manuscript.

20. P8 L12: Clarify that the test functions are acting as Lagrange multipliers.

We have added a statement about that in the second paragraph on page 8.

21. P8 Eq. 22: I would like to see a little bit more information about B. It is reasonably obvious how it can be constructed (i.e. after discretization evaluating the basis functions at the locations of the observation points). However, since the least-squares misfit isn't integrated over  $\Gamma_t$ , I fail to see how by equating it with something that is (namely the adjoint stress) can allow the strong form to be recovered. It seems to me that the authors are performing an additional integration of  $\Gamma_t$  that is not being shown in the paper, and I would like to see some additional specificity here.

We use pointwise observations independently from the mesh and as you observe, the operator requires the evaluation of basis functions at points. In Eq. 22,  $\mathcal{B}$  is the observation operator that maps the surface velocity field to a finite number of observation points.  $\mathcal{B}^*$  is the dual of  $\mathcal{B}$  which maps the velocity observations to the surface velocity field, i.e., given a vector w defined on the observation points,  $(\mathcal{B}^*w, v)_{\Gamma_t} = (w, \mathcal{B}v)$  for any function v defined on  $\Gamma_t$ . As a consequence,  $B^*w$  is a linear combination of Dirac delta functions, weighted by the components of w. After discretization, the observation operator is denoted by the matrix B, and  $B^* = B^T$ . We added the definition of  $\mathcal{B}^*$  to the paper.

22. P8 L30: Consider using underbraces to specify what part of the adjoint stress comes from where (and what gets omitted when using the linearized approximation). This is interesting because it becomes clear that there exists an additional term derived from the thermomechanically coupling term that I have never seen considered before.

We prefer to avoid underbraces, but we added a description of the terms (including a reference to our previous work highlighting the difference) after the definition of the adjoint stress.

P9 L3: Throughout the work, the authors use several different notational conventions to refer to effective strain-related quantities (i.e. \(\vec{\varepsilon}\) I, \(\vec{\varepsilon}\) i, tr(\(\vec{\varepsilon}^2\))). It would be better to use a consistent notation throughour (I would favor either the invariant or colon notation, but not the trace notation).

We have removed the trace notation, but for keeping expressions easy to read, we kept both the invariant and the colon notation. The former makes the formulation of Glen's law concise, and the latter is also used for the scalar product between other second-order tensors.

24. P9 L22: Without stating what this new functional is, I can't really determine if the computed variations are correct. Even referring back to Petra (2012), I can't tell what this Lagrangian on

the gradient referred to in the text actually looks like. Is it only the forward and adjoint equations imposed via new Lagrange multipliers, or is there a term to be minimized as well? It's not clear from the text.

The incremental equations can be computed in different ways. We changed the explanation to the more common way to derive these equations, namely by taking second variations of the Lagrangian (given in (17)). We have also added a reference to a textbook.

25. P10 L5: The incremental forward problem seems to also resemble the adjoint problem, particularly in the changes to the viscosity term relative to the self-adjoint case. Is there a more compelling reason why the forward and incremental forward problems should be more closely related than the forward and adjoint problems?

P10 L25: Same question as above.

This really depends on the problem under consideration. The incremental equations are linearizations (and adjoints of linearizations) of the forward equation, which partly explains why they look familiar.

26. P11 L7: Please make clearer from the outset that the content of this paragraph is paraphrased from Petra (2012), and that a more thorough discussion of the topic is included there.

We now explicity refer to Petra et al. for more details.

27. P11 L10: Can you cite a reference showing why neglecting terms involving the adjoint variable guarantees positive-definiteness in the Hessian? Petra (2012) did this as well, but similarly did not cite a reason for why this should be true.

We added a reference to a paper from Bangerth (2008) to our paper. Additionally, we also point to the book by Nocedal and Wright, who discuss the Gauss-Newton approximation for nonlinear least squares problems.

P11 L29: Check out the "citet" command in bibtex to get citations that look like "Isaac et al. (2015)", rather than "(Isaac et al., 2015)".

Corrected.

29. P12 L4-6: This front matter is redundant.

We disagree. A well written paper should tell the reader what to expect next and what the plan for an upcoming section is. For example, based on a reading of this material, the reader may decide to skip the section or come back to it later.

**30.** P12-13 Sec. 4.1: The specification of the finite element spaces splits the discussion about streamline upwinding. Consider transposing the first and second paragraphs.

We don't see where such a split is happening. In Section 4.1 we first explain the need for stabilization, we then define the discretized spaces needed for the SUPG-stabilized discretization of the forward problem, and we finally close the section with a discussion of the effect of the stabilization on the optimization problem (which is then further discussed in Section 4.2).

31. P13 L20: The discussion of Galerkin vs. non-Galerkin spaces requires a reference.

In Section 4.1 (toward the end) we have added a reference to a book by Mikhlin.

32. P14 L7: This point was already stated in the previous paragraph.

We changed the explanation to avoid that duplication.

**33.** P14 L13: Since the adjoint temperature isn't used here except as a means to facilitate optimization, is the lack of equivalency in the small mesh limit between the discrete and continuous (rather, non SUPG) temperature all that meaningful?

The inconsistency between the discrete and continuous adjoint temperature can degrade the convergence of the discrete adjoint temperature to the continuous adjoint temperature, and thus degrade the convergence of the discrete inverse solution to the continuous inverse solution. However, this is a subtle and technical detail.

34. P14 L17: This sentence provides no information.

It does provide some information, basically saying that there is not a clear winner. We have added a reference to Gunzburger's book, which discusses and compares the two appraoches in some detail in Section 2.9.

35. P14 L21-27: I'm not sure that this is true; the authors spent the previous paragraph talking about how the advantage of OTD is that one can use stabilization terms that vanish as the residual does, but that this creates other problems. This section would seem to suggest (wrongly) that it is DTO that has this property. I could of course be wrong, in which case the authors might humor me by re-writing this section more transparently.

We have made some minor modifications to this paragraph that will hopefully help to clarify this issue.

**36.** P14 L32: Perhaps "We consider both a two-dimensional flowline case and a three-dimensional map plane case" would be more clear.

We prefer to keep the "two- and three-dimensional" wording, which aligns well with the literature, including the benchmark paper by Pattyn, Perichon, Aschwanden, et al., 2008.

37. P14 L33: s(x) is usually reserved for surface elevations in glaciological parlance, where H(x) is reserved for ice thickness. In these examples the two appear to coincide because of the flat bed, but this could become confusing for other examples.

We have corrected this.

38. P14 L33: Why this particular choice of geometry? There are analytical solutions that are somewhat more realistic than a cosine, such as the Vialov profile. I don't think it matters much with respect to the results, but I think that the choice should be elaborated upon a little bit.

The choice of the geometry does not matter much. In reality, the ice sheet surface is irregular, which is determined by the accumulation rate. We did not choose the Vialov profile since it is based on the assumption that the accumulation rate is a positive constant over the entire domain, and all the ablation is by calving at the edge.

39. P16 L5: This could be explored with a little bit more depth. Throughout the remainder of the work it seems as though the methods presented herein are selecting overly smooth solutions. Could this be a result of using too aggressive regularization? Additionally,  $\gamma$  computed via Morozov's discrepancy principle would be different for each problem. Was it computed independently for each example, and how did it vary? Would using a smaller value of  $\gamma$  allow for better resolution of the small scale variability in geothermal heat flux or is the lack of detail indeed a result of the smoothing nature of Stokes' flow?

For each example, we use the Morozov discrepancy criterion (Vogel, 2002) to find a (near) optimal regularization parameter such that  $||u - u^{obs}|| \approx$  the noise level, where u is the solution computed

with various regularization parameters. Indeed, choosing regularization parameters that are too large would lead to an overly smoothed reconstruction, while a regularization parameter that is too small for the error in the observations would result in instability in the inverse problem, manifesting as noise in the inverse solution. We note that when  $G \approx G_{true}$ , the misfit is close to the noise level. Therefore, using this criterion, we make sure to avoid fitting the data more closely than warranted by the true solution (i.e., fitting the noise). The small-scale variations in the geothermal heat flux, as we explain on top of page 17, cannot be recovered due to the smoothing property of the thermo-mechanically coupled Stokes solution operator.

40. Figure 3: The chosen colormap is a little difficult to read. Try something higher contrast.

We wanted to choose a colormap that shows cold ice in blue and tempered ice in red tones, rather than a colormap like hsv that uses several different colors. Hence, we'd prefer keeping the currently used colormap.

**41.** P17 L2: Once again, I'm not sure whether these results are showing the limits of data recoverability or if they're showing the choice of regularization parameter. A bit more exploration of the latter topic would help to clear this up.

As we explain above, the regularization was carefully chosen (for each example) so that we don't fit the noise. This topic is further explored in the following paragraphs, where we systematically investigate the effect of the SNR on the inversion results.

42. Figure 6: I cannot tell the difference between cyan and blue in this figure.

We assume you refer to the right figure, where the cyan and black solid lines coincide. We now plot cyan on top of black and remark in the caption that the cyan line is on top of the black solid line.

43. P19 L6: I am extremely skeptical of the capacity for a model with 2 elements (quadratic or not) to accurately capture the vertical structure of the temperature field, particularly when said temperatures are subject to boundary layers (as the authors noted). Is it possible that this low vertical resolution is contributing to some of the error in the recovered solution? I suppose that since the same model is being used to generate the surface velocities that this effect would be lessened, but then that brings up the additional issue of whether the overly simplified temperature fields induced by the low resolution are leading to an easier job of recovering the geothermal heat field.

As now discussed at the beginning of Section 5.1, we have compared our *two-dimensional* results with results obtained on more refined meshes in the vertical direction. In particular, we have used 8 uniformly distributed layers of quadratic elements, as well as 6 non-uniform boundary-layer resolving layers of elements, and we found identical numerical results results.

For the *three-dimensional* results, we have replaced all numerical results using a finer mesh with *four* quadratic elements in the vertical direction. Going from two to four elements we found that the thermal boundary layers are resolved better. The optimal regularization parameter is the same as when the coarser mesh was used, and the number of GN iterations stayed the same. The upstream reconstruction is similar, but the downstream reconstruction worsened compared to the lower-resolution model. All results reported in the revised version of the paper now use the finer mesh. We appreciate your suggestions and comments.

We always add noise to the forward solution to generate the synthetic data. This is to lessen the "inverse crime".

44. Figure 9: Add legends to (a) and (b). Also, it would be useful to see an additional plot of the cosine of the angle between the two-way coupled descent direction and the exact gradient. Perhaps we would see some "uphill traffic" there as well.

We added the plot of the cosine of the angle between the search direction based on the exact (two-way coupled) gradient and the steepest descent direction (negative exact gradient). Since the Gauss-Newton approximation of the Hessian guarantees that the search direction is a descent direction, as expected, this new plot does not suggest any "uphill traffic".

45. Appendix: Probably not necessary to include this; a reference to any numerical analysis text ought to be fine.

We removed the Appendix section.

[revised manuscript text omitted]
},\boldsymbol{\theta}) := \frac{1}{2} A(\boldsymbol{\theta})^{-\frac{1}{n}} \left( \dot{\boldsymbol{\varepsilon}}_{\mathrm{II}} + \boldsymbol{\epsilon} \right)^{\frac{1-n}{2n}}$$
(5)

15 is bounded from below (Hutter, 1983; Jouvet and Rappaz, 2012).

The energy equation (3) is an advection-diffusion equation for the temperature field with a strain heating term on the right hand side. Note that the Stokes system (1, 2) and the energy equation (3) are two-way coupled: the velocity governed by the Stokes equations is the advection velocity in the energy equation and it additionally enters through the strain heating term on the right side of (3). In the opposite direction, the temperature enters in the Stokes equations through the viscosity term given in (4) and thus affects the flow field.

The domain  $\Omega$  is taken as a two- or three-dimensional ice slab with the following boundary conditions. On the top surface,  $\Gamma_t$ , we impose a traction-free boundary condition for the momentum equation and an imposed temperature for the energy equation. At the base of the ice sheet,  $\Gamma_b$ , we assume that the ice is below the pressure melting point and frozen to the bedrock. Hence, the boundary conditions are no-slip conditions for the momentum equation and thermal flux conditions for the energy equation representing the flux of geothermal heat into the ice from below (Greve and Blatter, 2009). Additional conditions for

the lateral boundaries for the model problems used in our study are specified in Section 5.

25

20

In summary, the forward problem is given by

$$\nabla \cdot \boldsymbol{u} = 0 \qquad \qquad \text{in } \Omega, \tag{6}$$

$$-\nabla \cdot \boldsymbol{\sigma}_{\boldsymbol{u}} = \rho \boldsymbol{g} \qquad \qquad \text{in } \Omega, \tag{7}$$

$$\rho c \boldsymbol{u} \cdot \nabla \theta - \nabla \cdot (K \nabla \theta) = 2\eta \operatorname{tr}(\dot{\boldsymbol{\varepsilon}}_{\boldsymbol{u}}^{2}) : \dot{\boldsymbol{\varepsilon}}_{\boldsymbol{u}} \qquad \text{in } \Omega,$$
(8)

$$\boldsymbol{\sigma}_{\boldsymbol{u}}\boldsymbol{n} = \boldsymbol{0}, \ \boldsymbol{\theta} = \boldsymbol{\theta}_s \qquad \qquad \text{on } \boldsymbol{\Gamma}_t, \tag{9}$$

$$\boldsymbol{u} = \boldsymbol{0}, \ K\nabla\theta \cdot \boldsymbol{n} = G \qquad \qquad \text{on } \boldsymbol{\Gamma}_b, \tag{10}$$

+ additional lateral B.C.s,

where n is the outward unit normal vector on  $\Gamma_t$  or  $\Gamma_b$ ,  $\theta_s$  is the prescribed temperature at the top surface, and G is the geothermal heat flux, and the expressions for the stress  $\sigma_u$  have been given previously.

10 Next, we present a weak form of the forward problem (6)–(10), which serves as the basis for the finite element discretization of these equations (Hughes, 2000), and is also used in the Lagrangian functional in Section 3. This weak form is found by multiplying the Stokes system (1, 2) and the energy equation (3) by test functions, integrating over  $\Omega$ , integrating by parts where appropriate and adding up the three weak equations. The weak form of the forward problem (6)–(10) is thus: Find  $(\boldsymbol{u}, p, \theta) \in \mathcal{U} \times \mathcal{P} \times \mathcal{T}$  such that

15
$$a(\boldsymbol{u}, p, \theta; \hat{\boldsymbol{v}}, \hat{q}, \hat{\lambda}) = \langle \rho \boldsymbol{g}, \hat{\boldsymbol{v}} \rangle_{\Omega} + \langle G, \hat{\lambda} \rangle_{\Gamma_b},$$
 (11)

for all test functions  $(\hat{v}, \hat{q}, \hat{\lambda}) \in \mathcal{U} \times \mathcal{P} \times \mathcal{T}_0$ , where

$$a(\boldsymbol{u}, p, \theta; \hat{\boldsymbol{v}}, \hat{q}, \hat{\lambda}) = \int_{\Omega} (2\eta(\boldsymbol{u}, \theta) \dot{\boldsymbol{\varepsilon}}_{\boldsymbol{u}} : \dot{\boldsymbol{\varepsilon}}_{\hat{\boldsymbol{v}}} - p\nabla \cdot \hat{\boldsymbol{v}} - \hat{q}\nabla \cdot \boldsymbol{u}) \, d\boldsymbol{x} + \int_{\Omega} (\rho c \hat{\lambda} \boldsymbol{u} \cdot \nabla \theta + K \nabla \theta \cdot \nabla \hat{\lambda}) \, d\boldsymbol{x} - \int_{\Omega} (2 \hat{\lambda} \eta(\boldsymbol{u}, \theta) \dot{\boldsymbol{\varepsilon}}_{\boldsymbol{u}} : \dot{\boldsymbol{\varepsilon}}_{\boldsymbol{u}}) \, d\boldsymbol{x},$$

and

25

5

$$\langle \rho \boldsymbol{g}, \hat{\boldsymbol{v}}
angle_{\Omega} = \int\limits_{\Omega} \rho \boldsymbol{g} \cdot \hat{\boldsymbol{v}} \, d\boldsymbol{x}, \text{ and } \langle G, \hat{\lambda}
angle_{\Gamma_b} = \int\limits_{\Gamma_b} \hat{\lambda} G \, d\boldsymbol{s}$$

20 Here,  $\dot{\varepsilon}_{\hat{v}}$  is the strain rate tensor based on  $\hat{v}$ , and ':' denotes the scalar product between second-order tensors. The spaces in the above equations are defined as

$$\mathcal{U} = \{ \boldsymbol{u} : \Omega \to \mathbb{R}^d \mid \boldsymbol{u}|_{\Gamma_b} = \boldsymbol{0} \},$$

$$\mathcal{P} = \{ \boldsymbol{p} : \Omega \to \mathbb{R} \},$$

$$\mathcal{T} = \{ \boldsymbol{\theta} : \Omega \to \mathbb{R} \mid \boldsymbol{\theta}|_{\Gamma_t} = \boldsymbol{\theta}_s \},$$

[revised manuscript text omitted]

functional. The Lagrangian functional  $\mathcal{L}$  combines the cost functional (13) with the weak form (11) of the forward problem, 25 with test functions  $\hat{v}$ ,  $\hat{q}$  and  $\hat{\lambda}_{\text{becoming}}$ . These test functions act as Lagrange multipliers and become the adjoint velocity v, adjoint pressure q, and adjoint temperature  $\lambda$ :

$$\mathcal{L}(\boldsymbol{u}, \boldsymbol{p}, \boldsymbol{\theta}; \, \boldsymbol{v}, \boldsymbol{q}, \lambda; \, \boldsymbol{G}) := \mathcal{J}(\boldsymbol{G}) + a(\boldsymbol{u}, \boldsymbol{p}, \boldsymbol{\theta}; \boldsymbol{v}, \boldsymbol{q}, \lambda) - \langle \boldsymbol{\rho} \boldsymbol{g}, \boldsymbol{v} \rangle_{\Omega} - \langle \boldsymbol{G}, \lambda \rangle_{\Gamma_b}.$$

$$\tag{17}$$

The gradient of  $\mathcal{J}$  with respect to the unknown heat flux G is found by requiring that variations of the Lagrangian  $\mathcal{L}$  with 30 respect to the forward and adjoint variables vanish. The gradient  $\mathcal{G}(G)$  is then found by taking the variation of  $\mathcal{L}$  with respect to G. In strong form, the gradient evaluated at  $G, \mathcal{G}(G)$ , is then given by:

$$\mathcal{G}(G) := \begin{cases} -\nabla \cdot (\gamma T \nabla G) - \lambda & \text{on } \Gamma_b, \\ (\gamma T \nabla G) \cdot \overline{n} & \text{on } \partial \Gamma_b, \end{cases}$$
(18)

[revised manuscript text omitted]